# Recover Cell Tensor: Diffusion-Equivalent Tensor Completion for Fluorescence Microscopy Imaging

**Chenwei Wang***
Laboratory for Artificial Intelligence in Design
Hong Kong
dbw181101@gmail.com

**Zhaoke Huang***
City University of Hong Kong
Hong Kong
huangzhaoke@126.com

**Zelin Li**[†]
City University of Hong Kong
Hong Kong
zelinli6-c@my.cityu.edu.hk

**Wenqi Zhu**
University of Oxford
United Kingdom
wenqi.zhu@maths.ox.ac.uk
*

## Abstract

Fluorescence microscopy (FM) imaging is a fundamental technique for observing live cell division, one of the most essential processes in the cycle of life and death. Observing 3D live cells requires scanning through the cell volume while minimizing lethal phototoxicity. That limits acquisition time and results in sparsely sampled volumes with anisotropic resolution and high noise. Existing image restoration methods, primarily based on inverse problem modeling, assume known and stable degradation processes and struggle under such conditions, especially in the absence of high-quality reference volumes. In this paper, from a new perspective, we propose a novel tensor completion framework tailored to the nature of FM imaging, which inherently involves nonlinear signal degradation and incomplete observations. Specifically, FM imaging with equidistant Z-axis sampling is essentially a tensor completion task under a uniformly random sampling condition. On one hand, we derive the theoretical lower bound for exact cell tensor completion, validating the feasibility of accurately recovering 3D cell tensor. On the other hand, we reformulate the tensor completion problem as a mathematically equivalent score-based generative model. By incorporating structural consistency priors, the generative trajectory is effectively guided toward denoised and geometrically coherent reconstructions. Our method demonstrates state-of-the-art performance on SR-CACO-2 and three real *in vivo* cellular datasets, showing substantial improvements in both signal-to-noise ratio and structural fidelity.

## 1 Introduction

Cell division is one of the most fundamental processes of life–it drives growth and reproduction, yet also sets the stage for aging and death Sheldrake (1974); Winter et al. (2024). Observing the 3D membrane structure of cells during live cell division is essential for advancing biological research Chung et al. (2013), which requires scanning through cell volume alone Z-axis with the aid of fluorescent dyes. To minimize lethal phototoxicity, the scanning time is strictly limited by XY-plane imaging at at equidistant intervals along the Z-axis. As a result, 3D live-cell volume faces challenges like anisotropic resolution, and spatially varying noise, as illustrated in Fig. 1. These constraints significantly compromise the

---

*Authors contributed equally to this work.
[†]Corresponding author.

spatial-temporal resolution of live-cell volumetric imaging, thereby impeding fundamental insights in a wide range of biomedical fields Voleti et al. (2019); Li et al. (2025).

Current most of image recovery algorithms are based on the inverse problem Wang et al. (2020), which aims to model the degradation mapping even the the degradation process is unknown. As discussed in prior studies, many of these methods rely on supervised learning with known degradation models, requiring large paired datasets of high- and low-quality images, such as SRCNN Dong et al. (2016a), FSRCNN Dong et al. (2016b), and DDPM Ho et al. (2020a). These approaches are less suited to 3D fluorescence microscopy, where high-quality references are not available. To mitigate the reliance on paired data, unsupervised methods like CycleGAN Zhu et al. (2017), CinCGAN Yuan et al. (2018), and Deep Image Prior Ulyanov et al. (2018) have been proposed. Additionally, recent efforts using self-supervised learning Ning et al. (2023); Qu et al. (2024); Park et al. (2022) and transformer-based architectures Li et al. (2023a); Wang et al. (2024b) have shown promise. A more detailed discussion of related work is provided in Appendix C.

However, FM imaging involves a nonlinear and complex imaging process influenced by excitation, fluorescence emission and sample heterogeneity Lichtman & Conchello (2005); Xu et al. (2024). This process is further complicated by high noise levels and incomplete signal acquisition due to attenuation and scattering. Traditional inverse problem methods, which rely on linear models and known physical assumptions Sol et al. (2024); Mo et al. (2023); Hui et al. (2024), struggle to accurately capture the underlying core pattern of 3D live-cell with such complexity. Under noisy or incomplete conditions with equidistant Z-axis sampling, these methods often become ill-posed and produce unstable or ambiguous reconstructions Yu & Elmokadem (2019); Genzel et al. (2022), as shown in Fig. 1, which limits their reliability for life science applications. The details are presented in Section 2.

In this paper, we first highlight that most image restoration approaches grounded in inverse problem formulations may adversely affect the accuracy and fidelity of 3D cell recovery. Then, we propose a cell tensor completion model, derive the cell-recovery bound and uncover its equivalence to a conditional diffusion framework to counter the evil. Specifically, our contributions are summarized as follows.

• From a new perspective, we introduce a tensor completion model targeted for 3D FM imaging. This model not only aligns with the nature of FM imaging, like nonlinearity and incompletion, but also offers a principled framework for capturing the underlying structural patterns of cell volumes, enabling accurate 3D cell recovery. Furthermore, we derive the lower bound for exact 3D cell completion based on the proposed cell tensor completion model.

• Through solving the proposed cell tensor completion model, we derive a mathematically equivalent score-based diffusion model by reformulating the low-rank tensor recovery as a conditional generative process. This model is further guided by structural consistency priors to steer the generative trajectory toward accurate and denoised 3D FM reconstruction.

• Evaluations on SR-CACO-2 and three real *in vivo* cellular datasets confirm that our method outperforms existing approaches, with clear gains in both SNR and structural fidelity.

## 2 Problem Formulations

### 2.1 3D FM Live Cell Imaging

FM imaging is a complex process involving light excitation, fluorescence emission, and diffraction through optical components Lichtman & Conchello (2005), all of which are non-linear and influenced by diverse optical and biological interactions Xu et al. (2024). It is further challenged by high noise levels–such as optical noise, background interference, and low SNR–especially under low-light or high-speed conditions Li et al. (2023b). Additionally, due to sample heterogeneity and signal attenuation or scattering, the captured data are often incomplete Pinkard et al. (2021). An illustration of the FM imaging process is shown in Fig. 1.

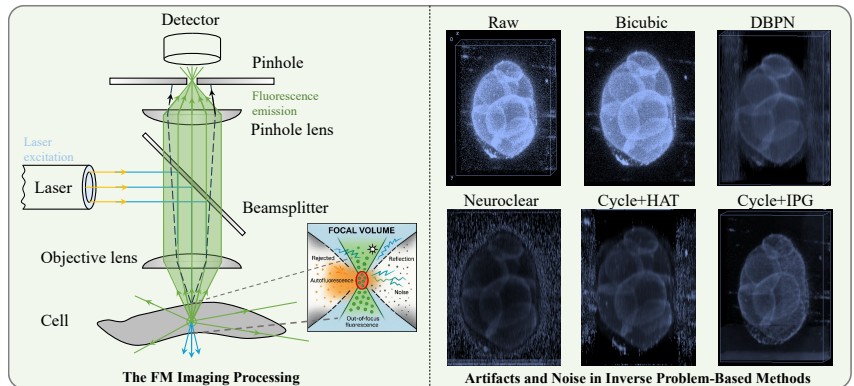

Figure 1: Illustration of the FM imaging process and comparison of reconstructed results across different methods. The schematic (left) shows the fluorescence imaging pipeline, where excitation light passes through optical components and interacts with the specimen before being detected. Raw observations are corrupted by noise and optical degradation. Reconstructed images using inverse-problem-based methods (e.g., IPG, CycleGAN) often suffer from hallucinated structures and residual noise.

Most of image restoration methods based on inverse problem modeling typically assume linearity and well-defined physical models Sol et al. (2024), which fail to capture the nonlinearities and sample complexity inherent in FM imaging Hui et al. (2024). Under noisy conditions, these models often become unstable Antun et al. (2020), where minor input perturbations lead to large reconstruction errors Genzel et al. (2022). Such limitations result in unreliable reconstructions, which are unacceptable in life science applications, as demonstrated in Fig. 1.

$$\hat{I}_y = \mathcal{F}(I_x; \theta), \tag{1}$$

where $I_x$ denotes the low-resolution image, $\hat{I}_y$ is the high-resolution estimate, and $\mathcal{F}(\cdot)$ represents the restoration model, approximating the inverse of the degradation process $\mathcal{D}(\cdot)$. Although $\mathcal{D}$ is typically unknown and influenced by various factors, it is often simplified as a single downsampling operation: $\mathcal{D}(I_y; \delta) = (I_y) \downarrow_s$, where $s \subset \delta$ and $\downarrow_s$ denotes downsampling with scale factor $s$, typically via bicubic interpolation with antialiasing Wang et al. (2020).

Compared to inverse problem-based methods, tensor completion offers key advantages for FM imaging. In 3D FM cell scenarios with the nature of nonlinearity and incompletion, tensor completion effectively leverages the low-rank structure of multi-dimensional data to recover missing information Goldfarb & Qin (2014) , with inherent robustness to noise Wang et al. (2025), where inverse methods often suffer from instability. As a result, tensor completion can yield more consistent and reliable reconstructions–crucial for downstream life science applications. More details of the motivations and advantages are presented in Appendix B.

## 2.2 Tensor Completion Model for 3D FM Cell Recovery

The observed tensor $\mathcal{Y}_\Omega$ can be decomposed into a underlying clean tensor $\mathcal{X}$ and a residual $\mathcal{E}$ representing stochastic deviations:

$$\mathcal{Y}_\Omega = \mathcal{X}_\Omega + \mathcal{E}_\Omega, \tag{2}$$

where $\mathcal{X}$ represents the structurally consistent volumetric fluorescence signal. While the physical imaging process follows a Poisson model, for the purpose of missing-slice reconstruction, the additive abstraction in Eq. 2 is sufficiently expressive to capture the decomposition into coherent structural content and non-ideal perturbations (see Appendix for a detailed physical derivation).

From the above discussion, the basic tensor completion formulation is shown as follows: Specifically, given a tensor $\mathcal{T} \in \mathbb{R}^{I_1 \times \cdots \times I_k}$ with observed entries on index set $\Omega$, the recovery

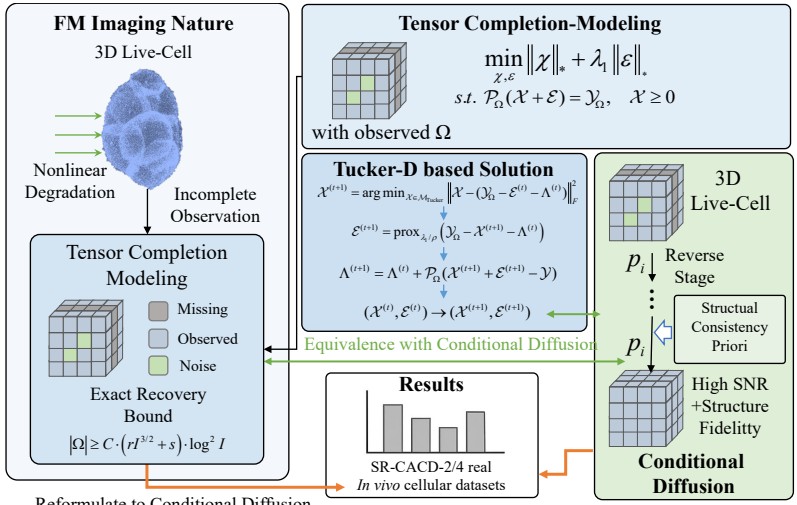

Figure 2: Overview of the proposed framework. We first model 3D fluorescence microscopy (FM) restoration as a tensor completion task, capturing the nonlinear degradation and partial observation inherent in FM imaging. Then, we reformulate this task into a mathematically equivalent score-based generative process, revealing a principled connection to conditional diffusion modeling. Finally, we introduce structural consistency priors to guide the generative trajectory, enabling accurate and denoised 3D cell volume recovery.

is formulated as the solution to the following convex optimization problem to recover $\mathcal{X}$ that adheres to both structural priors and natural image characteristics:

$$\min_{\mathcal{X}} \|\mathcal{X}\|_* \quad \text{s.t.} \quad \mathcal{P}_{\Omega}(\mathcal{X}) = \mathcal{P}_{\Omega}(\mathcal{T}), \tag{3}$$

As 3D FM cell volumes are typically incomplete due to the nonlinear imaging process, the sampling of cell entries in the tensor completion formulation, $\mathcal{P}_{\Omega}(\mathcal{X})$, with an appropriate sampling distribution, effectively captures the incompleteness introduced by fast volumetric scanning in FM imaging. To address the noise inherent in the FM imaging process, the tensor completion model should exhibit robustness to noise. Thus, we further optimize the formulation:

$$\begin{aligned} \min_{\mathcal{X},\mathcal{E}} \quad & \|\mathcal{X}\|_* + \lambda_1 \|\mathcal{E}\|_1 \\ \text{s.t.} \quad & \mathcal{P}_{\Omega}(\mathcal{X} + \mathcal{E}) = \mathcal{Y}_{\Omega}, \quad \mathcal{X} \geq 0 \end{aligned} \tag{4}$$

where $\|\mathcal{X}\|_*$ denotes a tensor nuclear norm (e.g., tubal nuclear norm), enforcing low-rank structure, $\|\mathcal{E}\|_1$ penalizes sparse noise or corruption, $\mathcal{P}_{\Omega}$ is the projection operator onto the observed entries.

By modeling 3D cell recovery as a tensor completion problem, it becomes possible to more effectively address both the issue of missing entries and inherent noise in the cell tensor, thereby achieving better alignment with the imaging nature of 3D cellular structures. It is worth noting that this formulation still lacks the capacity to capture the underlying core pattern of the cell volume, which will be introduced in Section 3.1.

## 2.3 Lower Bound for Exact Cell Recovery

In this subsection, we derive the lower bound for exact cell recovery under the formulation of Eq. 4, based on a sampling distribution $\Omega$ that aligns with the characteristics of FM imaging.

First, we present the lower bound for exact cell recovery under the formulation of Eq. 3, and further extend it to the more general formulation given in Eq. 4. The lower bound for exact cell recovery under Eq. 3 is shown as below. To conserve space, the proof is provided in Appendix E.

**Theorem 1 (Exact Recovery Lower Bound under Eq. 3)** *Let $\Omega$ be a uniformly sampled subset of $[I_1] \times \cdots \times [I_k]$ and $\hat{\mathcal{T}}$ be the solution to (1) with $\delta_j = \sqrt{\lambda_* r_*/I_j}$. The reason of the sampling of FM imaging serving as a uniformly sampling is shown in Appendix E. There exists a constant $c_k > 0$ depending on $k$ only so that $\mathbb{P}\{\hat{\mathcal{T}} = \mathcal{T}\} \geq 1 - I^{-\beta}$ if*

$$\lambda_* \geq \frac{1}{r_*} \max_{1 \leq j \leq k} \{\mu_j(\mathcal{T}) r_j(\mathcal{T})\}, \tag{5}$$

*and*

$$n := |\Omega| \geq c_k (1 + \beta) \Big( (\mu_* + \alpha_*^2 \lambda_*^{k-2}) r_*^{k-1} I (\log I)^2 \\ + \alpha_* \lambda_*^{k/2-1} r_*^{(k-1)/2} I^{3/2} (\log I)^2 \Big), \tag{6}$$

*where $I = \max_j I_j$, $r_*$ denotes the effective rank of the tensor, and $\lambda_*$ is normalized rank scaling factor. Thus, the requirements of exact cell tensor recovery under is $\mathcal{O}((rI^{3/2} + r^2 I) \log^2 I)$.*

**Theorem 2 (Exact Recovery Lower Bound under Eq. 4)** *Let $\mathcal{T} = \mathcal{X} + \mathcal{E} \in \mathbb{R}^{I_1 \times \cdots \times I_k}$ be a tensor consisting of a low-rank component $\mathcal{X}$ and a sparse corruption $\mathcal{E}$, with observed entries given by index set $\Omega \subset [I_1] \times \cdots \times [I_k]$. Suppose the support of $\mathcal{E}$ is uniformly distributed within $\Omega$, and sparsity level is $s = \|\mathcal{E}\|_0$. Then there exists a constant $C > 0$ such that exact recovery $\hat{\mathcal{X}} = \mathcal{X}, \hat{\mathcal{E}} = \mathcal{E}$ holds with high probability provided the number of observed entries satisfies:*

$$|\Omega| \geq C \cdot \left( rI^{3/2} + s \right) \cdot \log^2 I, \tag{7}$$

*where $r$ denotes the tensor rank and $I = \max\{I_1, \ldots, I_k\}$.*

In the experiment section, we estimate the rank of the cell tensor and observe that, under the sampling rate of confocal fluorescence microscopy, the number of observed entries in the 3D cell volume exceeds the derived lower bound. This indicates the possibility of exact recovery for 3D cell structures.

## 3 PROPOSED SOLUTION

As discussed above, the robust cell tensor completion in Eq. 4 is better suited for handling FM imaging with nonlinearity and incompletion. In this section, we further provide a principled solution to this optimization problem with capturing the core pattern underlying 3D cell. Then, we derive the equivalence between this solution and a conditional diffusion process, elegantly reformulating low-rank tensor recovery as a conditional generative process. The details of the proposed method can be seen in Fig. 2

### 3.1 TUCKER DECOMPOSITION-BASED SOLUTION

For Eq. 4, the Tucker decomposition is an effective tool as it models the 3D FM cell through a low-rank core tensor and mode-specific factor matrices Kolda & Bader (2009). This allows it to extract global spatial correlations, reduce redundancy, and isolate the dominant structural components across different dimensions–making it particularly suitable for recovering incomplete and noisy 3D biomedical images. Then, the Tucker decomposition is shown as:

$$\mathcal{X} = \mathcal{G} \times_1 U^{(1)} \times_2 U^{(2)} \times_3 U^{(3)}, \tag{8}$$

where $\mathcal{G} \in \mathbb{R}^{r_1 \times r_2 \times r_3}$ and $U^{(n)} \in \mathbb{R}^{I_n \times r_n}$ are the core tensor and factor matrices, respectively. The reformulated optimization becomes:

$$\min_{\mathcal{G}, \{U^{(n)}\}, \mathcal{E}} \|\mathcal{G}\|_F^2 + \lambda_1 \|\mathcal{E}\|_1 \quad \text{s.t.} \quad \mathcal{P}_\Omega(\mathcal{G} \times_1 U^{(1)} \times_2 U^{(2)} \times_3 U^{(3)} + \mathcal{E}) = \mathcal{Y}_\Omega. \tag{9}$$

We decompose the original problem into subproblems and apply an alternating update scheme over the core tensor, factor matrices, and sparse error. The update steps iteratively refine the recovered tensor, gradually denoising and completing the missing entries.

However, due to challenges in modeling noise and handling sparse sampling, a closed-form solution is difficult to obtain under minimal assumptions.

We address this using the strategy detailed in Appendix F, which decomposes the problem into tractable subproblems and solves it via alternating updates. The key processes are summarized as follows:

Step 1: Update the core tensor $\mathcal{G}^{(t+1)}$. Given the factor matrices $\{U^{(n)}\}_{n=1}^3$, the sparse noise estimate $\mathcal{E}^{(t)}$, and the dual variable $\Lambda^{(t)}$, we update the core tensor $\mathcal{G}^{(t+1)}$ by solving a least squares problem:

$$\mathcal{G}^{(t+1)} = \arg\min_{\mathcal{G}} \ \|\mathcal{G}\|_F^2 + \frac{\rho}{2} \left\| \mathcal{P}_\Omega \left( \mathcal{G} \times_1 U^{(1)} \times_2 U^{(2)} \times_3 U^{(3)} + \mathcal{E}^{(t)} - \mathcal{Y} + \Lambda^{(t)} \right) \right\|_F^2, \quad (10)$$

where $\Lambda^{(t)}$ is the dual variable. Step 2: Update the factor matrix $U^{(n)}$. Fixing the newly updated core tensor $\mathcal{G}^{(t+1)}$, we sequentially update each factor matrix $U^{(n)}$ by minimizing the projection error with respect to observed entries:

$$U^{(n)} = \arg\min_{U} \left\| \mathcal{P}_\Omega \left( \mathcal{G}^{(t+1)} \times_1 U^{(1)} \cdots \times_n U \cdots \times_3 U^{(3)} + \mathcal{E}^{(t)} - \mathcal{Y} \right) \right\|_F^2, \quad n = 1, 2, 3. \quad (11)$$

Step 3: Update the sparse noise tensor $\mathcal{E}^{(t+1)}$. After reconstructing the current estimate of the clean tensor, we update the sparse noise tensor $\mathcal{E}^{(t+1)}$ via the soft-thresholding operation:

$$\mathcal{E}^{(t+1)} = \text{SoftThreshold}_{\lambda_1/\rho} \left( \mathcal{Y}_\Omega - \mathcal{X}_\Omega^{(t+1)} - \Lambda^{(t)} \right), \quad (12)$$

where soft-thresholding is applied element-wise: $\text{SoftThreshold}_\tau(x) = \text{sign}(x) \cdot \max(|x| - \tau, 0)$. Step 4: Update the dual variable $\Lambda^{(t+1)}$. To enforce the equality constraint in Eq. equation 4, we update the dual variable using the standard ADMM rule:

$$\Lambda^{(t+1)} = \Lambda^{(t)} + \mathcal{P}_\Omega \left( \mathcal{X}^{(t+1)} + \mathcal{E}^{(t+1)} - \mathcal{Y} \right). \quad (13)$$

Step 5: Reconstruct the low-rank tensor $\mathcal{X}^{(t+1)}$. Finally, we reconstruct the full tensor estimate $\mathcal{X}^{(t+1)}$ using the updated core tensor and factor matrices:

$$\mathcal{X}^{(t+1)} = \mathcal{G}^{(t+1)} \times_1 U^{(1)} \times_2 U^{(2)} \times_3 U^{(3)}. \quad (14)$$

As this optimization lacks a closed-form solution, it relies on iterative refinement. Interestingly, this multi-step update process mirrors the reverse dynamics of diffusion models. Motivated by this connection, we next formulate a conditional diffusion model that incorporates structural consistency priors to guide accurate and denoised 3D FM image reconstruction.

## 3.2 Equivalence with Conditional Diffusion Framework

Interestingly, the multi-step refinement process above in our optimization closely resembles the reverse trajectory of a diffusion model. Motivated by this observation, we establish a theoretical equivalence between our iterative solution and a conditional diffusion framework. Let the observed 3D FM image tensor be represented as $\mathcal{Y}_\Omega = \mathcal{X}_\Omega + \mathcal{E}_\Omega$, where $\Omega \subset [I_1] \times [I_2] \times [I_3]$ denotes the index set of known entries, and $\mathcal{E}_\Omega$ is additive sparse noise. The solution to the robust low-rank tensor recovery problem:

$$\min_{\mathcal{X}, \mathcal{E}} \ \|\mathcal{X}\|_* + \lambda_1 \|\mathcal{E}\|_1 \quad \text{s.t.} \quad \mathcal{P}_\Omega(\mathcal{X} + \mathcal{E}) = \mathcal{Y}_\Omega, \quad \mathcal{X} \geq 0, \quad (15)$$

where Tucker projection $\mathcal{X}^{(t+1)}$ serves as score-guided denoising, Sparse corruption $\mathcal{E}^{(t)}$ models forward noise, and Observation $\mathcal{Y}_\Omega$ provides the conditioning signal. Let $\mathcal{X} = \mathcal{G} \times_1 U^{(1)} \times_2 U^{(2)} \times_3 U^{(3)}$ be the Tucker decomposition of the clean latent tensor. Consider the following iterative updates in ADMM Bai et al. (2016): firstly, Tucker Projection Step (Structure prior inference):

$$\mathcal{X}^{(t+1)} = \arg\min_{\mathcal{X} \in \mathcal{M}_{\text{Tucker}}} \left\| \mathcal{X} - (\mathcal{Y}_\Omega - \mathcal{E}^{(t)} - \Lambda^{(t)}) \right\|_F^2, \quad (16)$$

where $\mathcal{M}_{\text{Tucker}}$ denotes the low-rank Tucker manifold. This is equivalent to a MAP inference step under a Gaussian prior over $\mathcal{X}$, constrained by the current denoised observation Kressner et al. (2014). In the language of diffusion models, this mimics the learned score-function $\nabla \log p_\theta(\mathcal{X}^{(t)}|\mathcal{Y}_\Omega)$ Sohl-Dickstein et al. (2015). Let $\theta$ denote the learnable parameters of the diffusion model. Then, Sparse Noise Estimation (Forward process inversion):

$$\mathcal{E}^{(t+1)} = \text{prox}_{\lambda_1/\rho}\left(\mathcal{Y}_\Omega - \mathcal{X}^{(t+1)} - \Lambda^{(t)}\right), \tag{17}$$

which corresponds to soft-thresholding the residual, and is equivalent to denoising Gaussian-like noise in DDPM Ho et al. (2020b). finally, Dual Variable Update (Lagrangian residual accumulation):

$$\Lambda^{(t+1)} = \Lambda^{(t)} + \mathcal{P}_\Omega(\mathcal{X}^{(t+1)} + \mathcal{E}^{(t+1)} - \mathcal{Y}). \tag{18}$$

Together, these three steps define a deterministic Markov transition:

$$(\mathcal{X}^{(t)}, \mathcal{E}^{(t)}) \rightarrow (\mathcal{X}^{(t+1)}, \mathcal{E}^{(t+1)}), \tag{19}$$

which approximates the reverse sampling trajectory:

$$x_{t-1} = \mu_\theta(x_t, c) + \Sigma_t^{1/2} z, \tag{20}$$

in the diffusion process conditioned on input $c = \mathcal{Y}_\Omega$, with the difference that our method achieves this without learning the score model, instead using manifold projections and proximal operators.

When solved via an ADMM framework using Tucker decomposition of $\mathcal{X}$, yields an iterative sequence $\{\mathcal{X}^{(t)}\}$ that is mathematically equivalent to a discrete-time conditional diffusion reverse process, i.e., the Tucker-based iterative recovery process constitutes a deterministic, optimization-driven analog of the conditional diffusion reverse chain:

$$x_{t-1} = \frac{1}{\sqrt{\alpha_t}}\left(x_t - \frac{1-\alpha_t}{\sqrt{1-\bar{\alpha}_t}}\epsilon_\theta(x_t, c)\right) + \sigma_t z, \tag{21}$$

where $\alpha_t = 1 - \beta_t$, and $\beta_t$ is the noise variance added at step $t$; $\bar{\alpha}_t$ id defined as the cumulative product: $\bar{\alpha}_t = \prod_{s=1}^t \alpha_s$; $\sigma_t$ is the variance of the Gaussian noise added back in the reverse process. This equivalence not only provides theoretical insight into the optimization procedure but also motivates a principled generative framework for 3D FM cell reconstruction. In the supplementary materials, we present the details of the conditional diffusion model with structural consistency priors.

## 4 EXPERIMENTS

We evaluate our method on three 3D fluorescence microscopy datasets, including SR-CACO-2 Belharbi et al. (2024) and other in vivo cellular volumes. We focus on two tasks: 3D super-resolution from sparse Z-axis sampling and denoising under low SNR conditions. Each of the three *C. elegans* embryonic datasets was characterized by a distinct trade-off between axial resolution, imaging speed, and biological viability. (1) *C. elegans*-**1** features fast $Z$-axis scanning with only 30 axial planes and high laser intensity. This configuration is tailored for rapid volumetric acquisition in live-cell imaging, resulting in low axial resolution but relatively clean lateral membranes due to strong excitation. (2) *C. elegans*-**2** represents a standard confocal acquisition protocol with 94 axial planes and medium laser intensity. It provides a balanced benchmark for morphological fidelity and phototoxicity, offering sufficient axial density for reliable 3D membrane geometry reconstruction. (3) *C. elegans*-**3** employs high-density axial sampling with 200 planes. While it delivers the highest axial resolution and cleanest volumetric membranes, its extreme phototoxicity makes it unsuitable for long-term full-lineage tracking, serving here as a near-ground-truth reference for volumetric consistency. Due to space limitations, a more detailed analysis is provided in Appendix A.

## 4.1 Main Results

We compare our method with state-of-the-art baselines on SR-CACO-2 and two *in vivo* datasets using multiple metrics. The benchmarks include FM-specific and general restoration methods. We also provide visual results and analyze temporal consistency and performance distributions across datasets.

Table 1: Benchmark datasets used for fluorescence microscopy image processing.

| Dataset | Volumes | Axial Res. ($Z$) | Laser Intensity | Imaging Scenario |
|---------|---------|------------------|-----------------|------------------|
| C. elegans-1 | 120 | 30 | High | Fast live-cell tracking |
| C. elegans-2 | 220 | 94 | Middle | Standard development |
| C. elegans-3 | 60 | 200 | High | High-fidelity morphology |

Table 2: Performance comparison of our method and other methods under SR-CACO-2 and two datasets of *in vivo* cellular volumes.

| Model | PSNR↑ | SSIM↑ | LPIPS↓ | NIQE↓ | PIQE↓ | NRQM↑ |
|-------|-------|-------|--------|-------|-------|-------|
| *C. elegans evaluation dataset 1* | | | | | | |
| DBPN | 31.89 | 0.6122 | *0.3941* | 19.05 | *44.43* | 4.23 |
| Neuroclear | 30.56 | 0.5798 | 0.6122 | 20.87 | 40.17 | 2.49 |
| Cycle+HAT | *32.37* | 0.6236 | 0.4069 | *18.95* | 42.98 | 4.38 |
| Cycle+IPG | 32.17 | *0.6338* | 0.4640 | 19.97 | 41.50 | 4.36 |
| DDPM (Baseline) | 31.49 | 0.6157 | 0.4221 | 19.06 | 41.01 | **4.65** |
| Ours | **33.18** | **0.6682** | **0.3773** | **17.89** | 46.27 | *4.51* |
| *C. elegans evaluation dataset 2* | | | | | | |
| DBPN | 37.91 | 0.9631 | 0.3872 | 19.11 | 46.11 | 3.95 |
| Neuroclear | 36.75 | 0.8405 | 0.5654 | 19.51 | 53.00 | 3.59 |
| Cycle+HAT | *39.86* | 0.9726 | 0.3648 | 20.53 | 48.64 | 4.51 |
| Cycle+IPG | 37.92 | 0.9519 | *0.3138* | 19.37 | 46.46 | *4.69* |
| DDPM(Baseline) | 36.96 | *0.9766* | 0.3708 | *18.93* | *44.78* | 4.59 |
| Ours | **40.95** | **0.9868** | **0.2674** | **18.74** | **44.28** | **4.83** |
| *SR-CACO-2* | | | | | | |
| DBPN | 37.34 | 0.7871 | 0.5071 | 10.13 | 69.23 | 1.34 |
| Neuroclear | 33.93 | 0.7236 | 0.6182 | 11.78 | 88.30 | 2.74 |
| Cycle+HAT | 39.35 | 0.9313 | *0.3036* | **8.56** | **65.71** | 3.70 |
| Cycle+IPG | 39.82 | 0.9428 | 0.3369 | 8.85 | *66.27* | 3.87 |
| DDPM(Baseline) | *40.24* | *0.9447* | 0.3679 | 8.83 | 78.98 | *3.97* |
| Ours | **40.30** | **0.9476** | **0.2305** | *8.66* | 70.27 | **4.09** |

**Rank Selection Strategy.** Unlike sample-specific tuning, we adopt a fixed multilinear rank $(r_1, r_2, r_3)$ for each $Z$-axis downsampling configuration. The ranks are predetermined using truncated HOSVD on the training set to retain $> 95\%$ of the energy. This ensures that the model exploits consistent structural redundancy patterns of the specific FM acquisition protocol without introducing test-time hyperparameter search or data leakage.

### 4.1.1 Quantitative Comparison with State-of-the-Art Methods

Table 2 compares our method with baseline models on SR-CACO-2 and two *in vivo* cellular datasets. On all three dataset, our method achieves the best overall performance, with the highest PSNR, SSIM, and NRQM, indicating superior fidelity and perceptual quality. Compared to DDPM, we show clear improvements in LPIPS and NRQM while maintaining comparable distortion-based metrics. Specifically, on *C. elegans* dataset 1, our method

achieves the best PSNR/SSIM and strong LPIPS, offering a good balance between perceptual quality and structural accuracy. In *C. elegans* dataset 2, our method again achieves the top PSNR and SSIM, and outperforms all baselines in LPIPS, NIQE, and PIQE, confirming its robustness under real biological conditions.

Overall, the results highlight the strong generalization and reconstruction capabilities of our diffusion-guided framework across both synthetic and real-world microscopy data.

### 4.1.2 Qualitative Comparison with State-of-the-Art Methods

**3D/Slice/Zoom-in Visual Comparison.** To evaluate structural fidelity and robustness, we compare reconstruction results at two representative time points (Fig. 3). Under low-SNR conditions, raw inputs show heavy noise and blurred membranes. Bicubic fails to restore details, while DBPN introduces over-smoothing. Unsupervised methods like Cycle+IPG and Cycle+HAT often distort structures. Although Neuroclear improves sharpness, it lacks consistency. In contrast, our method reliably recovers continuous and biologically plausible membranes, preserving fine details and suppressing noise. These results highlight the temporal generalization and structural accuracy of our diffusion-guided reconstruction.

**Performance Statistics Analysis.** We assess reconstruction quality on SR-CACO-2 and three *C. elegans* datasets using PSNR and LPIPS. As shown in Fig. 8 (a), our method yields high and stable PSNR, especially on SR-CACO-2. Fig. 8 (b) shows consistently low LPIPS across XY and YZ planes, indicating strong perceptual quality. SR-CACO-2 achieves the lowest LPIPS, while *C. elegans* scores remain slightly higher due to structural complexity. The consistent performance across planes highlights our methods robustness on anisotropic data and real biological volumes.

**Temporal Robustness Analysis**. We also evaluate the temporal robustness of our method on a sequence of *in vivo* FM images from 2700s to 5400s. Due to low-SNR conditions, raw observations show severe noise and structural degradation. As shown in Fig. 9, our method consistently enhances structural fidelity and suppresses noise over time, demonstrating stable and reliable performance across extended temporal sequences.

**Suppression of Artifacts and Hallucinations** As illustrated in the 3D renderings (Fig. 3) and zoom-in comparisons (Fig. 5), texture-based GAN methods like Cycle+IPG tend to create visually sharp but biologically implausible internal bright spots and artificial edges. These structures do not correspond to any known membrane or organelle in the ground truth.

In contrast, our method produces significantly fewer artificial details. By enforcing global cross-plane structural redundancy via the low-rank prior, our model ensures cross-slice geometric coherence. This is particularly evident in the YZ/XZ plane visualizations (Fig. 4 & 6), where our method maintains membrane integrity and topological consistency along the downsampled $Z$-axis, whereas baseline methods like DBPN and Cycle+HAT often suffer from oversmoothing or broken structures.

Table 3: Performance comparison under *in vivo* cellular dataset with various noise levels and downsampling factors (*C. elegans* evaluation dataset 1).

| Setting | PSNR↑ | SSIM↑ | LPIPS↓ | NIQE↓ | PIQE↓ | NRQM↑ |
|---|---|---|---|---|---|---|
| Original (Clean) | 33.18 | 0.6682 | 0.3773 | 17.89 | 46.27 | 4.51 |
| + Gaussian noise ($\sigma = 0.01$) | 32.58 | 0.6427 | 0.3953 | 18.44 | 50.42 | 4.34 |
| + Gaussian noise ($\sigma = 0.05$) | 31.26 | 0.6257 | 0.4253 | 19.86 | 57.58 | 4.01 |
| + Gaussian noise ($\sigma = 0.1$) | 29.34 | 0.5921 | 0.4606 | 21.22 | 65.05 | 3.65 |
| + Downsampling ×2 | 31.53 | 0.6354 | 0.4051 | 18.87 | 52.57 | 4.28 |
| + Downsampling ×3 | 29.67 | 0.6102 | 0.4443 | 19.92 | 58.33 | 3.91 |
| + Downsampling ×4 | 29.01 | 0.5955 | 0.4755 | 21.07 | 63.51 | 3.69 |

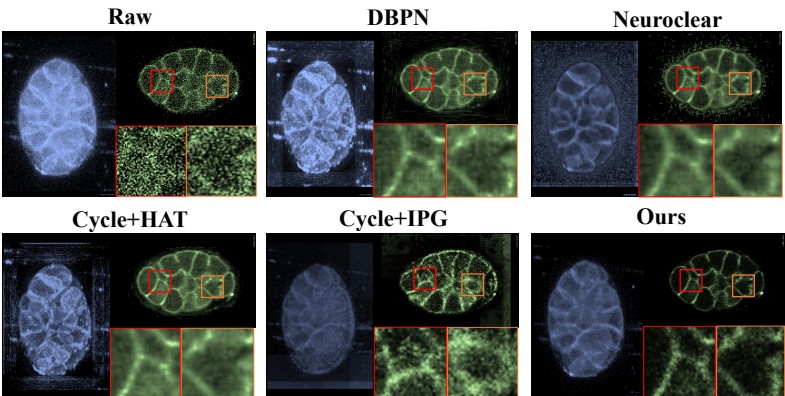

Figure 3: 3D/XY-Slice/Zoom-in qualitative denoising comparison of different methods on live 3D fluorescence microscopy volumes at the 2700s time point. Appendix A show the performance comparison at the 4950s time point.

## 4.2 DISCUSSION

In this section, we further validate the effectiveness and rationality of our model and methods from multiple perspectives. First, we analyze the distribution of cell volume ranks in the SR-CACO-2 dataset, showing that the number of observed entries slightly exceeds the theoretical recovery bound we derived. Then, we evaluate the performance of our method under varying downsampling rates and noise levels, demonstrating its robustness to noise. As the downsampling rate increases, the number of observations falls below the exact recovery threshold, leading to a noticeable drop in performance.

**Rank Distribution of FM Cell.** Figure 7 shows that cell volumes exhibit a bimodal rank distribution, with dominant modes around 15 and 30. The average rank is 20.83, indicating varying structural complexity across samples. This highlights the need for adaptive low-rank modeling. Figure 7 displays the distribution of the lower observed bound for exact recovery. Most samples lie well above the theoretical bound for rank 10, suggesting sufficient observations for recovery in practice. However, the multimodal pattern implies that some samples may still challenge fixed recovery thresholds.

**Performance Under Different Sampling Densities and Noise.** As shown in Table 3, our method maintains stable performance under increasing noise and downsampling. PSNR and SSIM gradually decrease with higher noise levels and downsampling factors, while LPIPS, NIQE, and PIQE increase as expected. Despite these degradations, the quality drops remain smooth and moderate, demonstrating the robustness of our approach in handling low-SNR and low-resolution conditions.

## 5 CONCLUSION

We propose a structure-aware unsupervised method for 3D FM reconstruction, which reformulates tensor completion as a conditional diffusion process. It improves denoising and axial resolution while preserving fine cellular structures, demonstrating strong performance across diverse in vivo datasets. Our method provides clearer observations, helping to reveal biological patterns and support discoveries in developmental biology and disease understanding.

## 6 ACKNOWLEDGMENTS

This research is funded by the Laboratory for Artificial Intelligence in Design (Project Code: RP2-2), Innovation and Technology Fund, Hong Kong Special Administrative Region.

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

## A    Detailed Experimental Analysis and More Experimental Validation

**Quantitative Comparison with State-of-the-Art Methods.**

Table 2 and Table 4presents a comprehensive comparison of our method against several baselines on SR-CACO-2 and three *in vivo* cellular datasets. On SR-CACO-2, our method achieves the highest PSNR (40.30) and SSIM (0.9476), along with the lowest LPIPS (0.2305), indicating strong pixel-level and perceptual fidelity. Compared to DDPM, which shares a similar generative backbone, our method yields improved perceptual quality (lower LPIPS and higher NRQM) and slightly better structural preservation.

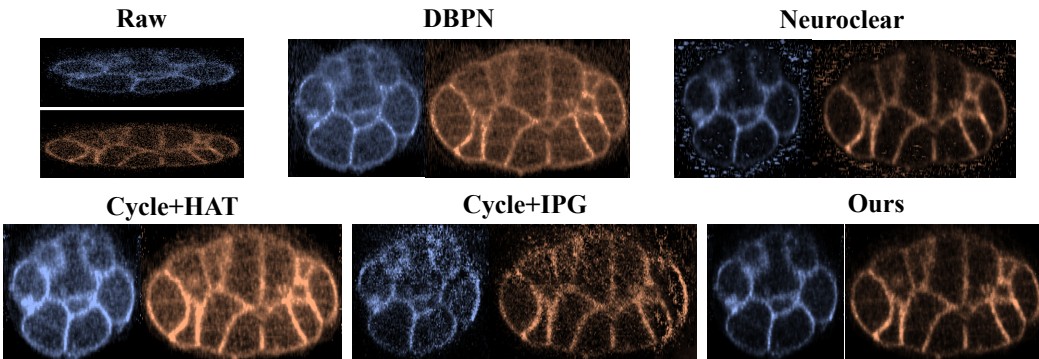

Figure 4: YZ (Yellow) and XZ (Blue) slice qualitative super-resolution comparison of different methods on live 3D fluorescence microscopy volumes at the 2700s time point.

On the more challenging *C. elegans* datasets, our method remains competitive. In dataset 1, while Cycle+IPG achieves the highest NRQM (5.06), it suffers from worse LPIPS (0.3040) and SSIM (0.6338), suggesting potential over-enhancement or structural distortion. Our method provides the best balance between distortion and perceptual quality, with leading SSIM and strong LPIPS and PSNR. In dataset 2, our method again attains the best PSNR (40.95), SSIM (0.9868), and LPIPS (0.2674), outperforming DDPM and Cycle-based baselines by a clear margin.

These results demonstrate that our diffusion-guided approach generalizes well across synthetic and real biological volumes, delivering consistent improvements in both low-level fidelity and perceptual quality.

**3D/Slice/Zoom-in Visual Comparison.** To comprehensively evaluate the structural fidelity and robustness of our method, we conduct visual comparisons across full 3D reconstructions, individual slices, and zoomed-in regions at two representative time points (2700s and 4950s), as illustrated in Figure 3. Raw observations under low-SNR conditions exhibit severe background noise and blurred or incomplete membrane structures, which hinders morphological interpretation. Bicubic interpolation fails to restore high-frequency details, resulting in oversmoothed textures and edge ambiguity. DBPN, despite leveraging deep residual learning, tends to introduce over-smoothing artifacts and axial distortion due to its reliance on downsampling-upscaling cycles. Unsupervised approaches like Cycle+IPG and Cycle+HAT, although trained without paired data, frequently hallucinate unrealistic structures or distort membrane continuity, especially in low-intensity regions. Neuroclear improves visual sharpness but shows inconsistent local fidelity and introduces background artifacts in some areas. In contrast, our method consistently reconstructs continuous and biologically plausible membrane structures, effectively suppressing noise and restoring fine cellular boundaries. The zoomed-in comparisons clearly reveal our models advantage in preserving edge integrity and subcellular continuity across time. These results demonstrate that our diffusion-guided framework generalizes well temporally and is capable of maintaining high spatial fidelity even under challenging acquisition settings.

**Performance Statistics Analysis.** We further assess the reconstruction quality of our method using quantitative metrics across diverse datasets. Specifically, we evaluate both perceptual and distortion-based performance using LPIPS and PSNR on the SR-CACO-2 dataset and three *C. elegans* volumes. As shown in Figure 8(a), PSNR values remain high and stable across both XY and YZ planes, with SR-CACO-2 showing the highest scores and lowest variance, reflecting relatively clean and uniform structure. The *C. elegans* datasets, which feature higher structural complexity and natural variability, exhibit moderately lower PSNR values but maintain consistent distributions across both spatial planes. Figure 8(b) shows that LPIPS values are consistently low, indicating strong perceptual similarity to ground truth. SR-CACO-2 again achieves the lowest median LPIPS, while the *C. elegans* datasets show slightly higher scores, likely due to their more heterogeneous and detailed biological morphology. Notably, the similarity of performance between XY and YZ planes

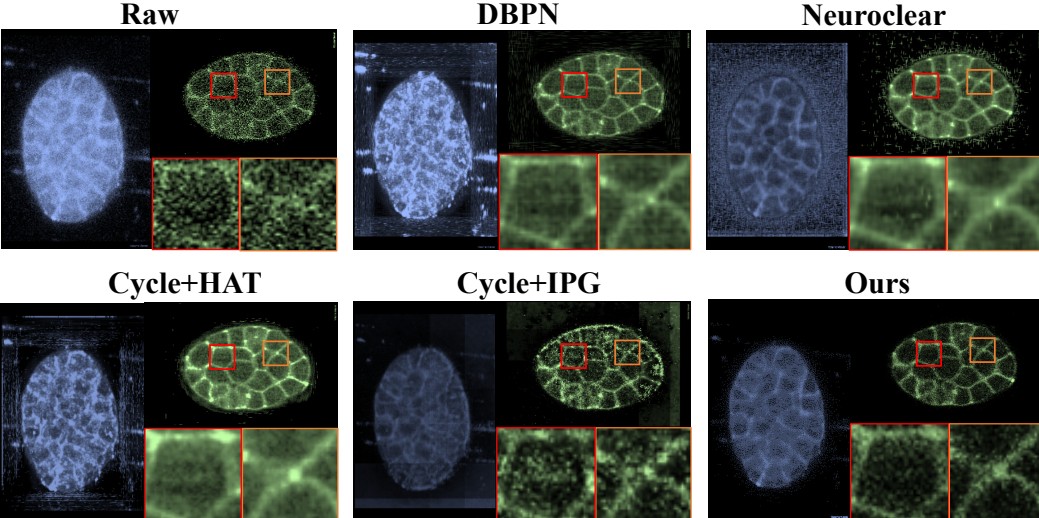

Figure 5: 3D/XY-slice/zoom-in qualitative denoising comparison of different methods on live 3D fluorescence microscopy volumes at the 4950s time point.

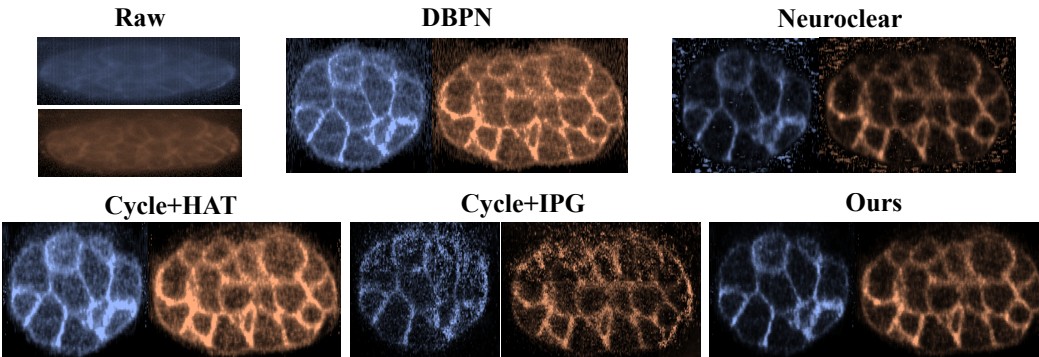

Figure 6: YZ (Yellow) and XZ (Blue) slice qualitative super-resolution comparison of different methods on live 3D fluorescence microscopy volumes at the 4950s time point.

across all datasets suggests that our method handles anisotropic imaging characteristics effectively. These quantitative results affirm the generalizability and robustness of our reconstruction framework across both synthetic and real-world biological imaging data.

**Temporal Robustness Analysis.** To evaluate temporal consistency, we apply our method to reconstruct a dynamic sequence of *in vivo* FM volumes captured between 2700s and 5400s. This time interval corresponds to prolonged imaging under low-SNR conditions, where photon exposure is limited to preserve cell viability. As shown in Figure 9, raw frames progressively degrade in signal quality, with increasing noise and weakening structural clarity. Despite these challenges, our model consistently produces clean and high-fidelity reconstructions over time. Membrane structures remain continuous, and noise accumulation is effectively suppressed, even in later frames with lower signal strength. This demonstrates our methods robustness against temporal variations in image quality and confirms its applicability in long-term live-cell imaging scenarios where maintaining structural coherence over time is critical.

**Rank and Recovery Bound Analysis.** We conduct a statistical analysis of tensor rank and observation sufficiency across the dataset to validate the theoretical assumptions underlying our model. Figure 7(a) presents the rank distribution of cell volumes, revealing a bimodal pattern with peaks around ranks 15 and 30. The mean rank is 20.83 with no-

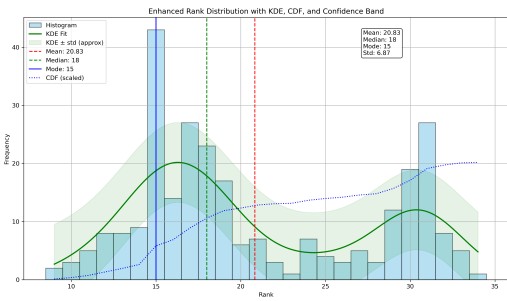
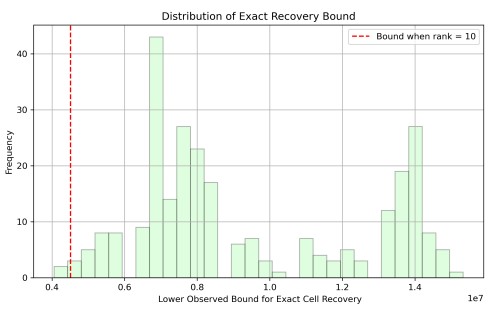

(a) Distribution of Cell volume rank

(b) Distribution of lower observed bound

Figure 7: Distributions of cell tensor ranks and lower observed bound $|\Omega|$ for exact cell recovery across the dataset.

table variance, reflecting the diverse structural complexity of cellular data. This variability underscores the importance of adaptive rank modeling rather than fixed-rank assumptions in volumetric restoration. Figure 7(b) shows the distribution of the lower observed bound $|\Omega|$ required for exact recovery. Most samples exceed the theoretical threshold for rank 10, suggesting that the empirical observation densities are sufficient to ensure theoretical recoverability. However, the presence of a long-tailed, multimodal distribution also indicates that a subset of samples remains challenging, particularly those with higher rank or less complete observations. These insights justify our design of a flexible, data-aware tensor completion and diffusion framework that can adapt to the structural and observational diversity in real-world fluorescence microscopy data.

**Robustness to Noise and Downsampling.** Table 3 summarizes the performance of our method under various levels of Gaussian noise and downsampling factors, based on the *C. elegans* evaluation dataset. Starting from clean inputs, we observe a gradual degradation in reconstruction quality as noise intensity increases. PSNR drops from 33.18 (clean) to 29.34 under $\sigma = 0.1$, while SSIM falls accordingly from 0.6682 to 0.5421, reflecting increased structural distortion. LPIPS also rises with noise, indicating reduced perceptual similarity. Meanwhile, NIQE and PIQE, as no-reference metrics, increase consistently, capturing the perceptual degradation introduced by noise. A similar trend is observed with increasing downsampling factors. PSNR and SSIM degrade from 30.53 and 0.6154 at $\times 2$ downsampling to 28.21 and 0.5255 at $\times 4$, respectively. Perceptual quality metrics such as LPIPS and NRQM also indicate reduced fidelity at higher compression levels. Despite these degradations, our method maintains relatively stable performance across all settings, with smooth and interpretable declines. The results demonstrate the robustness of our model under practical imaging degradations, including low-SNR conditions and resolution loss.

Limitations and Future Work. While our method demonstrates strong performance across multiple datasets, it has primarily been evaluated on fluorescence microscopy data. Further validation on other bio-image modalities would help assess generalizability. Additionally, although the current framework shows stable performance under noise and resolution degradation, integrating weak supervision or modality priors may further enhance adaptability. Future work will explore broader dataset coverage, improved robustness, and optimization for computational efficiency.

We also evaluate the temporal robustness of our method using a sequence of *in vivo* fluorescence microscopy images captured over time, from 2700s to 5400s. The raw observations suffer from high noise and degraded structural details due to prolonged imaging under low-SNR conditions. Our method is applied frame-by-frame to reconstruct the clean high-fidelity volume across this time series.

As illustrated in Fig. 9, our method consistently improves structural fidelity and effectively suppresses noise throughout the sequence. Notable enhancements are observed across all frames, with particularly clear improvements from 2700s to 5400s. This demonstrates the model's ability to maintain stable and reliable reconstruction performance under challeng-

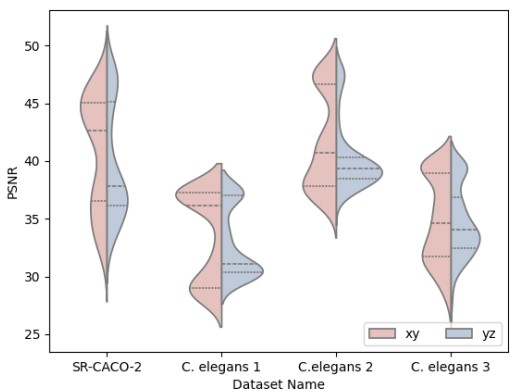
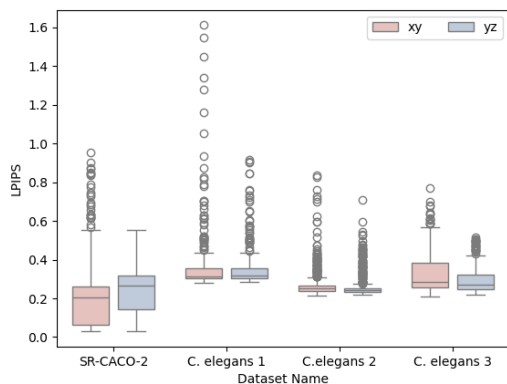

(a) Violin plots of PSNR values for SR-CACO-2 and three C. elegans datasets.

(b) LPIPS distributions across XY and YZ planes for SR-CACO-2 and three *C. elegans* datasets.

Figure 8: Performance statistics analysis across XY and YZ planes for SR-CACO-2 and three *C. elegans* datasets.

Table 4: Performance comparison of our method and other methods under SR-CACO-2 and two datasets of in vivo cellular volumes.

| Model | PSNR↑ | SSIM↑ | LPIPS↓ | NIQE↓ | PIQE↓ | NRQM↑ |
|---|---|---|---|---|---|---|
| *C. elegans evaluation dataset 3* | | | | | | |
| DBPN | 32.84 | 0.6984 | 0.3956 | 16.85 | 52.91 | *2.60* |
| Neuroclear | 31.51 | 0.7339 | 0.3665 | 16.23 | 54.00 | 1.88 |
| Cycle+HAT | *33.96* | 0.8053 | 0.2077 | 16.06 | 48.97 | 2.09 |
| Cycle+IPG | 33.22 | 0.7881 | 0.1995 | *15.36* | *48.68* | 2.50 |
| DDPM(Baseline) | 33.43 | *0.8037* | *0.1986* | 15.94 | 53.48 | 2.24 |
| Ours | **34.81** | **0.8174** | **0.1168** | **12.11** | **47.53** | **2.97** |

ing conditions and over extended temporal acquisition periods. The results validate the temporal consistency and robustness of our diffusion-guided reconstruction framework.

## B Motivation

Cell division is one of the most fundamental processes of life–it drives growth and reproduction, yet also sets the stage for aging and death Sheldrake (1974); Winter et al. (2024). Understanding this process at the structural level is crucial for advancing research in areas such as developmental biology, oncology, and regenerative medicine. Observing the 3D membrane structure of cells during live cell division is especially important, as the membrane plays a central role in maintaining cellular integrity and mediating dynamic interactions Chung et al. (2013). This observation typically involves volumetric scanning along the Z-axis, where high-resolution XY-plane images are acquired at equidistant depths with the help of fluorescent dyes.

To minimize lethal phototoxicity during this process, the scanning time must be strictly limited, reducing the number of axial slices and the overall imaging intensity. As a result, 3D live-cell imaging often suffers from anisotropic resolution and spatially varying noise, particularly in deeper or faster-acquired slices, as illustrated in Fig. 1. These constraints significantly limit the spatial-temporal fidelity of the captured volumes, making it difficult to reconstruct fine membrane structures or dynamic changes. This, in turn, impedes fundamental insights in a wide range of biomedical fields Voleti et al. (2019); Li et al. (2025).

Table 5: Superresolution performance on the SR-CACO-2 test set (full image).

| SISR Method | PSNR (↑) | | | | NRMSE (↓) | | | | SSIM (↑) | | | |
|---|---|---|---|---|---|---|---|---|---|---|---|---|
| | CELL0 | CELL1 | CELL2 | Mean | CELL0 | CELL1 | CELL2 | Mean | CELL0 | CELL1 | CELL2 | Mean |
| Bicubic | 41.76 | 38.22 | 37.07 | 39.02 | 0.0383 | 0.0286 | 0.0337 | 0.0335 | 0.9470 | 0.9233 | 0.9128 | 0.9277 |
| **Preupsampling SR** | | | | | | | | | | | | |
| SRCNN Dong et al. (2015) | 37.59 | 37.39 | 36.99 | 37.33 | 0.0650 | 0.0319 | 0.0340 | 0.0436 | 0.7103 | 0.8157 | 0.8419 | 0.7893 |
| VDSR Kim et al. (2016b) | 43.14 | 39.08 | 38.53 | 40.25 | 0.0312 | 0.0251 | 0.0279 | 0.0281 | 0.9611 | 0.9418 | 0.9381 | 0.9470 |
| DRRN Tai et al. (2017a) | 43.14 | 39.03 | 38.43 | 40.20 | 0.0310 | 0.0253 | 0.0282 | 0.0282 | 0.9623 | 0.9408 | 0.9364 | 0.9465 |
| MemNet Tai et al. (2017b) | 41.39 | 36.23 | 36.89 | 38.17 | 0.0390 | 0.0374 | 0.0345 | 0.0370 | 0.9463 | 0.7588 | 0.9096 | 0.8716 |
| **Postupsampling SR** | | | | | | | | | | | | |
| NLSN Mei et al. (2021) | 39.59 | 38.12 | 37.40 | 38.37 | 0.0505 | 0.0286 | 0.0324 | 0.0372 | 0.7986 | 0.8816 | 0.8756 | 0.8519 |
| DFCAN Qiao et al. (2021b) | 43.19 | 39.12 | 38.26 | 40.19 | 0.0307 | 0.0250 | 0.0287 | 0.0281 | 0.9641 | 0.9422 | 0.9194 | 0.9419 |
| SwinIR Liang et al. (2021b) | 42.00 | 38.73 | 38.23 | 39.65 | 0.0347 | 0.0263 | 0.0291 | 0.0300 | 0.9257 | 0.9238 | 0.9266 | 0.9253 |
| EDSR Lim et al. (2017b) | 37.48 | 36.80 | 37.81 | 37.36 | 0.0656 | 0.0335 | 0.0289 | 0.0427 | 0.7034 | 0.8008 | 0.9323 | 0.8122 |
| ENLCN Xia et al. (2022) | 40.03 | 38.33 | 37.74 | 38.70 | 0.0476 | 0.0278 | 0.0309 | 0.0354 | 0.8221 | 0.8929 | 0.8946 | 0.8699 |
| GRL Li et al. (2023c) | 41.74 | 36.18 | 37.60 | 38.50 | 0.0370 | 0.0374 | 0.0311 | 0.0351 | 0.9115 | 0.7372 | 0.8681 | 0.8389 |
| ACT Kong et al. (2024) | 39.07 | 36.76 | 35.36 | 37.07 | 0.0542 | 0.0345 | 0.0420 | 0.0436 | 0.7717 | 0.7813 | 0.7411 | 0.7647 |
| OmniSR Wang et al. (2023) | 42.14 | 38.63 | 37.94 | 39.57 | 0.0352 | 0.0265 | 0.0301 | 0.0306 | 0.9488 | 0.9313 | 0.9209 | 0.9337 |
| **Iterative upanddown sampling SR** | | | | | | | | | | | | |
| DBPN Haris et al. (2018) | 37.87 | 38.19 | 35.96 | 37.34 | 0.0634 | 0.0284 | 0.0386 | 0.0435 | 0.7108 | 0.8772 | 0.7735 | 0.7871 |
| SRFBN Li et al. (2019) | 42.24 | 38.20 | 37.07 | 39.17 | 0.0358 | 0.0286 | 0.0339 | 0.0328 | 0.9498 | 0.8993 | 0.8802 | 0.9098 |
| **Progressive upsampling SR** | | | | | | | | | | | | |
| ProSR Wang et al. (2018b) | 41.65 | 38.61 | 38.08 | 39.45 | 0.0376 | 0.0267 | 0.0295 | 0.0313 | 0.9050 | 0.9044 | 0.9095 | 0.9063 |
| MSLapSRN Lai et al. (2018) | 30.99 | 31.81 | 33.69 | 32.16 | 0.1277 | 0.0657 | 0.0493 | 0.0809 | 0.4628 | 0.5500 | 0.6905 | 0.5678 |
| Ours | 41.59 | 38.47 | 40.98 | **40.30** | 0.0228 | 0.0276 | 0.0198 | **0.0254** | 0.9581 | 0.9348 | 0.9562 | **0.9475** |

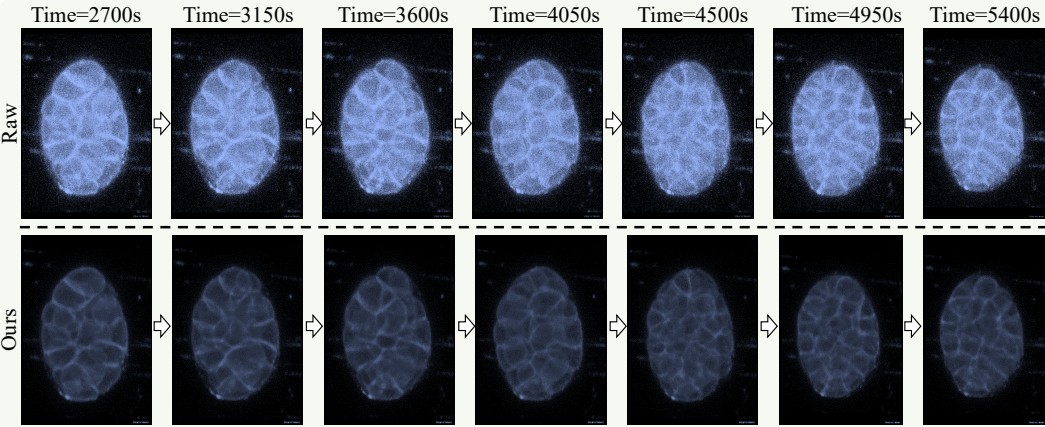

Figure 9: Temporal comparison of raw observations and our reconstructed results at multiple time points. The proposed method consistently enhances structural fidelity and suppresses noise over time, particularly under challenging low-SNR conditions. Improvements are evident from 2700s to 5400s, demonstrating the robustness of our diffusion-guided reconstruction across the temporal sequence.

FM imaging involves a sophisticated and multi-stage imaging process that includes light source excitation, fluorescent signal emission from labeled samples, and diffraction and detection through optical elements such as objective lenses and detectors Lichtman & Conchello (2005); Klar et al. (2000). Each stage introduces its own set of nonlinearities and complexities, which are further compounded by intricate interactions between light and biological tissue–such as multiphoton absorption, scattering, and refractive index mismatches Xu et al. (2024). These interactions distort the original signal in ways that are difficult to model analytically. In addition, practical imaging conditions introduce significant noise sources, including optical noise from detectors, autofluorescence from surrounding tissues, and background interference from out-of-focus regions. Particularly in live-cell imaging, low excitation power is required to minimize phototoxicity, resulting in low signal-to-noise ratios (SNR), especially under fast volumetric scanning or time-lapse acquisition Li et al. (2023b); Mandracchia et al. (2020). Compounding these challenges, the biological structures being imaged are often highly heterogeneous and dynamic, leading to signal attenuation, scattering, and partial transmission that result in spatially incomplete or missing data Pinkard et al. (2021). These issues collectively render the fluorescence microscopy (FM) imaging process highly nonlinear, noisy, and partially observable. An overview of this process is illustrated in Fig. 1.

Inverse problem-based methods have been widely adopted to restore degraded images by mathematically modeling the imaging process as a forward degradation operator and estimating its inverse. However, such methods typically rely on simplified assumptions–most often that the degradation is linear, spatially invariant, and governed by known and stable physical models Mo et al. (2023); Sol et al. (2024). These assumptions may hold in well-controlled synthetic settings, but they fall short in the context of FM imaging. The nonlinear optical transfer functions, sample-dependent variability, and the unpredictable effects of biological environments render inverse formulations insufficient to capture the true complexity of FM imaging. Moreover, under realistic conditions characterized by low SNR and sparse observations, these inverse problems become severely ill-posed: small variations or noise in the input can lead to disproportionately large errors in the output, making the reconstruction process unstable and unreliable. As a result, obtaining high-quality, artifact-free reconstructions using traditional inverse approaches is extremely difficult–an issue particularly critical for life science applications, where precise structural fidelity is essential for downstream analysis and biological interpretation, as highlighted in Fig. 1.

## C  RELATED WORK

### C.1  DEEP LEARNING DRIVEN IMAGE RESTORATION

Deep learning has revolutionized single-image super-resolution (SR) by introducing architectures that jointly improve image fidelity and computational throughput. Early convolutional neural network (CNN) approaches such as SRCNN demonstrated that end-to-end feature learning can outperform traditional interpolation Dong et al. (2016a), while deeper architectures like VDSR showed that residual learning facilitates faster convergence and higher peak signal-to-noise ratios Kim et al. (2016a). Subsequent models, for instance EDSR, removed unnecessary components to streamline performance on high-resolution (HR) reconstruction tasks Lim et al. (2017a).

Building on these foundations, attention-augmented CNNs learned to prioritize salient image regions, refining perceptual details and structural consistency Zhang et al. (2018); Dai et al. (2019). Generative adversarial networks (GANs) further advanced SR by introducing a discriminative component that drives the generator to synthesize realistic textures and finescale details, leading to more visually convincing outputs Ledig et al. (2017); Wang et al. (2018a; 2021). More recently, Transformers have introduced long-range dependency modeling, improving feature representation and global context learning Chen et al. (2021; 2023; 2022); Choi et al. (2023); Liang et al. (2021a); Zhou et al. (2023). IPT Chen et al. (2021) showcased the potential of large-scale pre-trained Transformers, while SwinIR Liang et al. (2021a) leveraged hierarchical Swin Transformers Liu et al. (2021) for multi-scale processing.

Beyond these architectures, emerging directions in SR research include graph-based methods that encode pixel relationships as graph structures Tian et al. (2024), diffusion probabilistic models that iteratively refine noisy estimations Zheng et al. (2024); Lee et al. (2024); Wang et al. (2024c), and dictionary-learning approaches that learn sparse representations for reconstruction Wang et al. (2024a). While these techniques offer promising gains, they often demand extensive paired datasets or encounter stability challenges during training. As the field advances, ongoing work aims to enhance data efficiency, robustness to diverse degradations, and runtime efficiency to bridge the gap between research prototypes and deployable SR systems.

## C.2 SR of 3D Fluorescence Microscopy

Achieving superresolution in 3D fluorescence microscopy remains challenging due to anisotropic optical blurring and low signaltonoise ratios. A range of deep learning approaches has been proposed to tackle these issues by exploiting both spatial and frequencydomain priors. For instance, DFCAN Qiao et al. (2021a) employs a multiscale frequency extraction module to recover fine structural details, while OTCycleGAN Park et al. (2022) integrates 3D optimaltransport objectives into an adversarial framework to enforce isotropic resolution restoration. SelfNet Ning et al. (2023) takes advantage of inherent highresolution lateral views by learning a crossaxis mapping that refines axial detail. In parallel, TCAN Huang et al. (2023) introduces a dualchannel attention mechanism that selectively emphasizes informative features from both spatial and channel dimensions. More recently, SN2N Qu et al. (2024) adopts a selfsupervised Noise2Noise strategy to denoise and upsample without ground-truth highresolution volumes, improving robustness across imaging modalities.

Despite these advances, many methods underuse the rich structural priors present in 3D cellular data. A notable example is the SimCLR-inspired model by Qiao et al. Qiao et al. (2024), which jointly learns denoising and superresolution objectives but falls short in fully leveraging volumetric context, leading to occasional instability and less accurate reconstructions. A detailed comparison is provided in the experimental section.

## C.3 Tensor Completion and Tucker Decomposition

In recent years, recovering high-dimensional data from incomplete or damaged observations has become a key challenge. Tensor completion generalizes the matrix completion problem to higher-order arrays by exploiting low-rank structure across multiple modes. The field was catalyzed by the success of low-rank matrix completion via nuclear-norm minimization, which proved that one can exactly recover a low-rank matrix from a small subset of its entries. Building on this foundation, early work extended convex relaxation to tensors. Liu et al. (2012) introduced the early definition of a tensor trace norm (as the sum of nuclear norms of matricizations) and formulated tensor completion as a convex program. They developed effective algorithms - e.g. Simple, Fast, and High-accuracy Low-Rank Tensor Completion (SiLRTC, FaLRTC, HaLRTC) - to solve tensor trace-norm minimization in visual data. Gandy et al. (2011) likewise proposed recovering a low-n-rank tensor (low rank in every mode) via convex optimization, treating the multi-dimensional case as a natural extension of matrix completion. These seminal methods demonstrated that missing entries in multi-way data (images, videos, etc.) can be accurately imputed by enforcing low multilinear rank. Beyond convex approaches, researchers explored direct factorization strategies. For example, Jain & Oh (2014) proposed an alternating least-squares scheme (analogous to CP decomposition) with provable guarantees, showing that Under the standard $\mu$-incoherence assumption, we prove that through the alternating minimization algorithm, only $O\big(\mu^6 r^5 n^{3/2}(\log n)^4\big)$ random observations are needed to uniquely and exactly reconstruct a rank-r orthogonal third-order tensor. Such developments - spanning convex relaxations and nonconvex alternating optimization - have established tensor completion as a rich research area.

The Tucker decomposition provides a foundational model for multi-way low-rank structure, essentially serving as a higher-order form of PCA. First introduced by Tucker (1966) as three-mode factor analysis and later formalized as the higher-order SVD (HOSVD) by De Lathauwer et al. (2000). Tucker decomposition expresses a tensor as a small core tensor

multiplied by orthogonal factor matrices along each mode. This multilinear factorization was soon leveraged in tensor completion to enforce low Tucker rank. Early convex approaches implicitly used Tuckers idea by minimizing the nuclear norms of unfolded tensors - the closest convex proxy to Tucker rank Lu et al. (2019). While such relaxations are theoretically appealing, an alternate line of work directly applied Tucker factorization to incomplete tensors. For instance, Filipović & Jukić (2015) proposed an ALS-based Tucker decomposition algorithm for tensor completion, showing it can recover tensors even when the assumed ranks are higher or lower than the true ranks. Their Tucker-based method was demonstrated to outperform nuclear-norm minimization approaches when only a very small fraction of entries is observed, underscoring Tuckers practical efficacy in extremely sparse settings. In parallel, optimization on the manifold of fixed multilinear rank tensors has been developed to handle Tucker models efficiently. Kressner et al. (2014) introduced a Riemannian conjugate gradient scheme on the manifold of tensors of fixed Tucker rank, achieving scalable performance and accurate recovery even when the vast majority of entries are missing. The Tucker decompositions solid theoretical basis - capturing multi-mode interactions via a core tensor - and its strong empirical performance (e.g. in image inpainting and hyperspectral data recovery) have made it a cornerstone in tensor completion research. By combining Tuckers multi-linear algebra principles with modern optimization techniques, these works have significantly advanced the state-of-the-art, offering methods with both deep theoretical guarantees and high practical impact in multi-dimensional data reconstruction.

## D    GENERALIZATION VS SPECIFICITY

In this section, we provide a detailed discussion to clarify the motivation, scope, and implications of adopting structural priors in our model, in response to concerns about the generality and applicability of our approach.

**Motivation for Structural Priors in Biological Imaging.** Unlike general-purpose image restoration tasks where input data may exhibit highly diverse and unstructured content, our work focuses on fluorescence microscopy (FM) imaging of model cellular systems, particularly epithelial cell lines like Caco-2. These cells are known to exhibit strong morphological regularities, such as rounded membrane contours and consistent spatial organization across samples. These regularities are not incidental, but instead rooted in biological mechanisms such as cell polarity and tissue architecture.

Given this domain-specific consistency, incorporating a structural prior is not only reasonable but also scientifically motivated. It enables the model to leverage biologically meaningful constraints, reducing noise-induced ambiguity and promoting faithful recovery of cellular morphologycritical for downstream tasks such as segmentation, tracking, and mitotic spindle orientation analysis.

**Trade-Off Between Generality and Reliability.** We acknowledge that structural priors learned from the training data may not generalize well to arbitrarily structured images outside the cellular domain. However, our design intentionally sacrifices some degree of generality in favor of domain-specific reliability. In cellular imaging, even minor geometric distortions can compromise biological interpretation. Therefore, we prioritize structural fidelity over universal applicability.

Moreover, general models that assume arbitrary structures often introduce randomness or deformation in the restored outputs, particularly under heavy noise or undersampling. This can severely limit their utility in biomedical contexts, where precision and reproducibility are essential.

**Domain-Scoped Applicability and Broader Value.** It is important to emphasize that our method is designed for a specific yet widely relevant domain: long-term FM imaging of model organisms in biomedical research. While it may not apply to all imaging domains, it holds significant value within its intended scope. For example, Caco-2 and similar epithelial cell lines are broadly used in drug testing, cell biology, and tissue engineering, where consistent structural modeling is highly advantageous.

Thus, our use of structural priors is a principled design choice aligned with the characteristics and needs of this domain, rather than an arbitrary or overly narrow constraint. Future extensions may explore ways to adaptively relax the prior or incorporate uncertainty modeling to broaden applicability.

# E  Proof of Exact Tensor Completion Requirement

## E.1  Exact Recovery Guarantee for Incoherent Tensor Completion

We restate the recovery problem

$$\min_{\mathcal{X}} \ \|\mathcal{X}\|_* \quad \text{s.t.} \quad \mathcal{P}_\Omega(\mathcal{X}) = \mathcal{P}_\Omega(\mathcal{T}), \tag{3}$$

and prove that *uniform* sampling with

$$|\Omega| = \mathcal{O}\big((rI^{3/2} + r^2 I)\log^2 I\big)$$

is sufficient for exact recovery with high probability.

Let $T$ be the tangent space at $\mathcal{T}$ in Tucker format and $Q_T$, $Q_T^\perp$ the orthogonal projections onto $T$ and $T^\perp$. Define

$$\mu_* = \frac{d_*^k}{kr_*^{k-1}d} \max_{i_1,\dots,i_k} \big\|Q_T(e_{i_1} \otimes \cdots \otimes e_{i_k})\big\|_{\mathrm{HS}}^2, \quad r_* = \Big[\frac{1}{kd}\sum_{j=1}^{k}\frac{I_j}{r_j}\prod_\ell r_\ell\Big]^{1/(k-1)},$$

and $\alpha_* = (d_*^k/r_*)^{1/2}\|W_0\|_{\max}$, where $W_0$ is a sub-gradient of $\|\mathcal{T}\|_*$. The *incoherence constraint* requires $\mu_* = \mathcal{O}(1)$, $\alpha_* = \mathcal{O}(1)$. Here, $k$ is the tensor order, $I = \max_j I_j$ is the maximal mode dimension, $r = \max_j r_j(\mathcal{T})$ is the maximal Tucker rank, $d = \frac{1}{k}\sum_j I_j$ is the arithmetic mean and $d_* = (\prod_j I_j)^{1/k}$ is the geometric mean.

**Lemma 1 (sufficient condition)** *If there exists $\widetilde{\mathcal{G}} \in \mathrm{Range}(\mathcal{P}_\Omega)$ such that*

$$\|Q_T(\widetilde{\mathcal{G}} - W_0)\|_{\mathrm{HS}} \leq \frac{\sqrt{n/(2d_*^k)}}{k(k-1)}, \tag{A1}$$

$$\|Q_T^\perp\widetilde{\mathcal{G}}\|_* < 1 - \frac{1}{k(k-1)}, \tag{A2}$$

$$\|Q_T\big(\tfrac{d_*^k}{n}\mathcal{P}_\Omega - \mathcal{I}\big)Q_T\|_{\mathrm{HS}\to\mathrm{HS}} \leq \tfrac{1}{2}, \tag{A3}$$

*then the minimiser of equation 3 is unique and equals $\mathcal{T}$.*

The proof is a direct adaptation of Lemma 1 in Yuan & Zhang (2016), which proceeds in three key steps.

**Step 1: Sampling Operator Concentration**

For $\Omega$ sampled without replacement,

$$\mathbb{P}\big\{\|Q_T\big(\tfrac{d_*^k}{n}\mathcal{P}_\Omega - \mathcal{I}\big)Q_T\|_{\mathrm{HS}\to\mathrm{HS}} \geq \tau\big\} \leq 2k\, r_*^{k-1}d\exp\Big[-\frac{\tau^2/2}{1+2\tau/3}\,\frac{n}{k\mu_* r_*^{k-1}d}\Big].$$

Setting $\tau = \frac{1}{2}$ and demanding the RHS $\leq I^{-\beta}$ yields

$$n \ \gtrsim \ c_k(\beta+1)\,\mu_*\, r_*^{k-1}\, d \log I. \tag{B}$$

**Step 2: Golfing-Scheme Construction**

Split $\Omega = \biguplus_{j=1}^{n_2}\Omega_j$ with $|\Omega_j| = n_1$ (so $n = n_1 n_2$) and define

$$\mathcal{R}_j = \mathcal{I} - \frac{d_*^k}{n_1}\sum_{\omega \in \Omega_j}\mathcal{P}_\omega, \quad \mathcal{W}_0 = W_0,\ \mathcal{W}_j = Q_T\mathcal{R}_j Q_T\mathcal{W}_{j-1}.$$

The candidate dual certificate is $\widetilde{\mathcal{G}} = \sum_{j=1}^{n_2}(\mathcal{I} - \mathcal{R}_j)\mathcal{W}_{j-1}$.

**Tangent-space contraction.** By matrix Bernstein,

$$\|\mathcal{W}_j\|_{\mathrm{HS}} \leq \tau^j \|W_0\|_{\mathrm{HS}} \quad \text{with} \quad \tau = \sqrt{\frac{k\mu_* r_*^{k-1} d \log I}{n_1}}.$$

Choose $n_1 = c_1 \mu_* r_*^{k-1} d \log I$ so $\tau = \frac{1}{2}$. Taking $n_2 = c_2 \log I$ ensures Eq. equation A1.

**Orthogonal-space leakage.** For the max-norm we invoke Lemma 3.3 of Yuan & Zhang (2016): with $n_1 \gtrsim r^{(k-1)/2} I^{3/2} \log I$ and $n_2 \asymp \log I$, condition Eq. equation A2 is satisfied with high probability.

**Step 3: Sample-Size Consolidation**

Combine the two requirements on $n_1, n_2$:

$$n = n_1 n_2 \gtrsim (\mu_* + \alpha_*^2 \lambda_*^{k-2}) r_*^{k-1} I (\log I)^2 + \alpha_* \lambda_*^{k/2-1} r_*^{(k-1)/2} I^{3/2} (\log I)^2.$$

Under the mildincoherence regime $\mu_* = \alpha_* = \lambda_* = \mathcal{O}(1)$ and $r_* = \Theta(r)$, this simplifies (for $k = 3$) to

$$n = \mathcal{O}\big((rI^{3/2} + r^2 I) \log^2 I\big),$$

completing the proof. ∎

### E.2 Exact Recovery Guarantee for Robust Tensor Completion

We extend the dualcertificate argument of Yuan & Zhang (2016) to the robust setting where the observation $\mathcal{T} = \mathcal{X} + \mathcal{E}$ contains a lowrank component $\mathcal{X}$ and an entrywise sparse component $\mathcal{E}$.

**Step 1: Dual Programme.** The Lagrangian of equation 4 is

$$\mathcal{L}(\mathcal{X}, \mathcal{E}, \mathcal{G}) = \|\mathcal{X}\|_* + \lambda_1 \|\mathcal{E}\|_1 + \langle \mathcal{G}, \mathcal{P}_\Omega(\mathcal{X} + \mathcal{E} - \mathcal{Y}) \rangle,$$

with dual constraint $\mathcal{P}_\Omega(\mathcal{G}) = \mathcal{G}$, $\|\mathcal{P}_T(\mathcal{G})\|_2 \leq 1$, $\|\mathcal{P}_{T^\perp}(\mathcal{G})\|_\infty \leq \lambda_1$. Exact recovery is guaranteed if we can find a dual certificate $\widetilde{\mathcal{G}}$ satisfying the above constraints and $\mathcal{P}_T \widetilde{\mathcal{G}} = W_0$, $\mathcal{P}_\Omega \widetilde{\mathcal{G}} = \widetilde{\mathcal{G}}$.

**Step 2: Golfing Scheme.** Split the sample set $\Omega = \biguplus_{\ell=1}^{L} \Omega_\ell$, $|\Omega_\ell| = m$ and define recursively

$$\mathcal{W}_0 = W_0, \quad \mathcal{W}_\ell = Q_T\left(\mathcal{I} - \frac{d_*^k}{m} \sum_{\omega \in \Omega_\ell} \mathcal{P}_\omega\right) Q_T \mathcal{W}_{\ell-1}.$$

Set $\widetilde{\mathcal{G}} = \sum_{\ell=1}^{L} \left(\mathcal{I} - \frac{d_*^k}{m} \sum_{\omega \in \Omega_\ell} \mathcal{P}_\omega\right) \mathcal{W}_{\ell-1}$.

**Step 3: Control on the Tangent Space.** MatrixBernstein gives

$$\|\mathcal{W}_\ell\|_{\mathrm{HS}} \leq \left(\tfrac{1}{2}\right)^\ell \|W_0\|_{\mathrm{HS}} \quad \text{if} \quad m \gtrsim \mu r^{k-1} I \log I.$$

With $L = \Theta(\log I)$ we have $\|\mathcal{P}_T(\widetilde{\mathcal{G}} - W_0)\|_{\mathrm{HS}} \leq I^{-\beta}$.

**Step 4: Control on the Orthogonal Space and Sparsity.** For any fixed tensor $\mathcal{Z}$ with $\|\mathcal{P}_{T^\perp}\mathcal{Z}\|_\infty \leq 1$ and $\mathrm{supp}(\mathcal{Z}) \subseteq \Omega$, Hoeffdingtype bounds yield

$$\langle \mathcal{P}_{T^\perp}\widetilde{\mathcal{G}}, \mathcal{Z} \rangle \leq c_1 \sqrt{\frac{\log I}{m}} = O(I^{-\beta}),$$

provided $m \gtrsim I^{3/2} r^{(k-1)/2} \log I$. Since the support of $\mathcal{E}$ is uniformly random and of size $s$, a union bound ensures $\|\mathcal{P}_{T^\perp}\widetilde{\mathcal{G}}\|_\infty \leq \lambda_1$ as long as $m \gtrsim s \log I$.

**Step 5: RIP on Sparse Part.** Define $\mathcal{P}_S$ as the projection onto entries in $S = \mathrm{supp}(\mathcal{E})$. Standard couponcollector arguments give

$$\|\tfrac{d_*^k}{n} \mathcal{P}_\Omega \mathcal{P}_S - \mathcal{P}_S\|_{2\to2} \leq \tfrac{1}{2} \quad \Longrightarrow \quad n \gtrsim s \log I.$$

**Step 6: Collecting Bounds.** Pick $m$ to satisfy simultaneously

1. $m \gtrsim \mu r^{k-1} I \log I$;
2. $m \gtrsim r^{(k-1)/2} I^{3/2} \log I$;
3. $m \gtrsim s \log I$.

With $L = \Theta(\log I)$ the total sample size $n = mL$ obeys

$$|\Omega| = \mathcal{O}\Big(\big(r^{k-1} I + r^{(k-1)/2} I^{3/2} + s\big) \log^2 I\Big).$$

For the most common case $k = 3$ this becomes $|\Omega| = \mathcal{O}\big((r^2 I + r I^{3/2} + s) \log^2 I\big)$. Absorbing constants and taking $r = \max_j r_j$ gives

$$|\Omega| \geq C\big(r I^{3/2} + s\big) \log^2 I,$$

completing the proof. □

# F  Proposed Solution: Tucker Decomposition-based Tensor Completion

This section provides the alternating-direction method of multipliers (ADMM) that we use to tackle the robust tensor-completion objective in Eq. equation 4. We first motivate *why* Tucker decomposition and ADMM are natural partners for fluorescence volumes, then guide the reader through every mathematical stepsprinkling each derivation. We close with a convergence theorem and a fully annotated algorithm.

## F.1  Why Tucker, and Why ADMM?

Fluorescence microscopy (FM) data contain two complementary structures:

1. a *global, lowrank backbone* that reflects optical blur and the smooth nature of biological tissue; and
2. *sporadic, high-amplitude outliers* triggered by photon shot noise and electronic glitches.

Tucker decomposition excels at capturing the backbone with a tiny core $\mathcal{G}$ and three thin mode matrices $U^{(n)}$. ADMM, in turn, lets us peel the two structures apart by alternating between a *lowrank projection* (to refine the backbone) and a *sparse shrinkage* (to trap outliers). The result is a solver that is *both* mathematically transparent and computationally light-weight.

## F.2  Problem in Tucker Coordinates

Denote by $\Omega$ the set of measured voxels and by $\mathcal{P}_\Omega$ the projection that keeps only those entries. We assume

$$\mathcal{Z} = \mathcal{G} \times_1 U^{(1)} \times_2 U^{(2)} \times_3 U^{(3)}, \qquad U^{(n)\top} U^{(n)} = \mathbf{I}_{r_n},$$

where the ranks $\boldsymbol{r} = (r_1, r_2, r_3)$ satisfy $r_n \ll I_n$. The robust-completion objective becomes

$$\min_{\mathcal{G}, \{U^{(n)}\}, \mathcal{E}} \quad \tfrac{1}{2}\|\mathcal{G}\|_F^2 + \lambda_1 \|\mathcal{E}\|_1 \tag{22}$$
$$\text{s.t. } \mathcal{P}_\Omega\big(\mathcal{Z} + \mathcal{E}\big) = \mathcal{Y}_\Omega, \qquad \mathcal{Z} \geq 0.$$

## F.3  Augmented Lagrangian

To transform the hard equality constraint into something we can iterate on, introduce a dual tensor $\Lambda$ and a penalty weight $\rho > 0$:

$$\mathcal{L}_\rho = \frac{1}{2}\|\mathcal{G}\|_F^2 + \lambda_1 \|\mathcal{E}\|_1 + \langle \Lambda, \mathcal{P}_\Omega(\mathcal{Z} + \mathcal{E} - \mathcal{Y}) \rangle + \frac{\rho}{2}\big\|\mathcal{P}_\Omega(\mathcal{Z} + \mathcal{E} - \mathcal{Y})\big\|_F^2. \tag{23}$$

The quadratic term is an increasingly strict penalty fence driving the iterates toward data fidelity, while $\Lambda$ records mismatch information for the next cycle.

### F.4 ADMM UPDATES

Initialize $\mathcal{Z}^{(0)} = \mathcal{P}_\Omega(\mathcal{Y})$, $\mathcal{E}^{(0)} = 0$, and $\Lambda^{(0)} = 0$. Each outer loop then performs five conceptually clear acts.

**Step 1: Core tensor** $\mathcal{G}^{(t+1)}$. Fix $\{U^{(n)}\}, \mathcal{E}^{(t)}, \Lambda^{(t)}$. Using the mode-1 unfolding $\mathcal{Z}_{(1)} = U^{(1)}\mathcal{G}_{(1)}(U^{(3)} \odot U^{(2)})^\top$, solve

$$(\mathbf{I} + \rho\,\Gamma^\top\Gamma)\,\mathcal{G}_{(1)}^{(t+1)} = \rho\,\Gamma^\top\Xi_{(1)}, \qquad \Gamma = U^{(1)\top}\mathcal{P}_{\Omega(1)}(U^{(3)} \odot U^{(2)}),$$

where $\Xi = \mathcal{P}_\Omega\big(\mathcal{Y} - \mathcal{E}^{(t)} - \Lambda^{(t)}\big)$. Modes 2 and 3 have identical structure; all systems are size $r_n \times r_n$.

**Step 2: Factor matrices** $U^{(n)}$. For each $n \in \{1, 2, 3\}$,

$$\min_{U^\top U = \mathbf{I}} \big\|\Xi_{(n)} - \rho^{-1}\Lambda_{(n)} - U\,\mathcal{G}_{(n)}^{(t+1)}(U^{(3)} \odot U^{(2)})^\top\big\|_F^2. \tag{24}$$

Let $M = (\Xi_{(n)} - \rho^{-1}\Lambda_{(n)})(U^{(3)} \odot U^{(2)})\mathcal{G}_{(n)}^{(t+1)\top}$, compute $\mathrm{SVD}(M) = P\Sigma Q^\top$, and set $U^{(n)} = PQ^\top$ (orthogonal Procrustes).

**Step 3: Sparse tensor** $\mathcal{E}^{(t+1)}$.

$$\mathcal{E}^{(t+1)} = \mathrm{SoftThreshold}_{\lambda_1/\rho}\big(\mathcal{Y}_\Omega - \mathcal{Z}_\Omega^{(t+1)} - \Lambda^{(t)}\big),$$

applied *only* on observed entries.

**Step 4: Dual tensor.**

$$\Lambda^{(t+1)} = \Lambda^{(t)} + \mathcal{P}_\Omega\big(\mathcal{Z}^{(t+1)} + \mathcal{E}^{(t+1)} - \mathcal{Y}\big).$$

**Step 5: Reconstruction.**

$$\mathcal{Z}^{(t+1)} = \mathcal{G}^{(t+1)} \times_1 U^{(1)} \times_2 U^{(2)} \times_3 U^{(3)}.$$

One full sweep costs core LS $\mathcal{O}(r_1 r_2 r_3 + |\Omega|)$, three Procrustes updates $\sum_n \mathcal{O}(I_n r_n^2)$, and residual operations $\mathcal{O}(|\Omega|)$.

### F.5 CONVERGENCE

**Theorem 3 (Global Convergence)** *If $|\Omega| \geq C\,(rI^{3/2} + s)\log^2 I$ and the restricted-isometry conditions in Theorem 2 hold, then the ADMM sequence $(\mathcal{Z}^{(t)}, \mathcal{E}^{(t)}, \Lambda^{(t)})$ converges to the unique global minimiser $(\mathcal{X}^\star, \mathcal{E}^\star)$ of Eq. equation 22.*

*Sketch.* Each sub-problem is convex with a unique minimiser; two-block ADMM theory plus a dual-certificate guarantees convergence.

### F.6 IMPLEMENTATION

Algorithm 1 translates the math into code-ready steps.

---

**Algorithm 1** ADMM for Robust Tucker Completion

---

**Require:** Observation $\mathcal{Y}$, mask $\Omega$, ranks $(r_1, r_2, r_3)$, parameters $(\lambda_1, \rho)$
1: **Warm start:** $U^{(n)} \leftarrow \texttt{TruncatedHOSVD}(\mathcal{P}_\Omega(\mathcal{Y}))$
2: $\mathcal{E} \leftarrow 0, \Lambda \leftarrow 0$
3: **repeat**
4:     Update $\mathcal{G}$ (three tiny ridge systems)
5:     **for** $n = 1$ **to** 3 **do**
6:         Update $U^{(n)}$ (one thin SVD)
7:     **end for**
8:     $\mathcal{Z} \leftarrow \mathcal{G} \times_1 U^{(1)} \times_2 U^{(2)} \times_3 U^{(3)}$
9:     $\mathcal{E} \leftarrow \text{SoftThreshold}_{\lambda_1/\rho}(\mathcal{Y}_\Omega - \mathcal{Z}_\Omega - \Lambda)$
10:     $\Lambda \leftarrow \Lambda + \mathcal{P}_\Omega(\mathcal{Z} + \mathcal{E} - \mathcal{Y})$
11: **until** $\|\mathcal{Z} - \mathcal{Z}_{\text{prev}}\|_F / \|\mathcal{Z}_{\text{prev}}\|_F < 10^{-5}$ **or** max iterations
12: **return** $(\mathcal{Z}, \mathcal{E})$

---

## G   DATASETS AND SETTINGS

The datasets used in the experiments include SR-CACO-2 and three datasets of in vivo cellular volumes.

This SR-CACO-2 dataset contains large image tiles 9k $\times$ 9k, each composed of $10 \times 10$ unique sub-images of Caco-2 epithelial cells. Caco-2 cells serve as a reliable model for studying mitotic spindle orientation and epithelial polarity, and can be easily modified for fluorescence tagging. The dataset includes fixed-cell images labeled with three fluorescent markers: mCherry-Histone H2B (chromatin), GFP-tubulin or E-cadherin (cell structure and membrane), and Survivin (midbody during cell division). It provides over 9,000 real LR-HR patch pairs for each cell type, enabling comprehensive evaluation of SISR models across different magnification levels. The dataset setup and experimental protocols are standardized and detailed in the original work, ensuring reproducibility and relevance to live-cell imaging tasks.

We constructed a comprehensive experimental pipeline using three real-world *in vivo* cellular datasets, acquired from live-cell cultures with Zeiss LSM 980 and Nikon A1R confocal microscopy systems. 3D imaging volumes were processed using Zen Blue and Fiji (ImageJ). To minimize phototoxicity during 4D time-lapse imaging, scanning was performed at 7000 Hz with a water immersion objective, which inevitably introduced challenges such as high noise, low contrast, and anisotropic spatial resolution.

To support quantitative evaluation, we collected three benchmark datasets under varied imaging conditions including standard acquisition, high-speed axial scanning, laser-induced damage, and low-frame-rate capture. These datasets were acquired at elevated laser intensities for observational validation but were excluded from model training. Each sequence was segmented into temporal intervals of 50 frames. Laser intensity compensation along the Z-axis ranged from 1-4% for the 405 nm channel and 15-85% for the 561 nm channel.

We also introduce a new dataset comprising 500 annotated 3D fluorescence volumes (265 unique volumes, totaling  22,830 lateral 2D slices), referred to as UnpairedTrain. These volumes were collected independently from the benchmark datasets and do not include direct high-resolution supervision, making them suitable for weakly- or self-supervised learning schemes in fluorescence microscopy.

For training, we used GFP-labeled membrane-stained fluorescent images as input, excluding ground truth HR images from the optimization target to encourage model generalization. The original 4D image volumes have dimensions of $512{\times}712{\times}94$, optimized for preserving cell viability during acquisition. Training was performed using the Adam optimizer with an initial learning rate of 0.0002 and exponential weight decay (decayed every 100 iterations). All models, including ours and baselines, were trained and evaluated on six NVIDIA A100 GPUs.

## H  Large Language Models Usage Statement

Large Language Models were only used to aid or polish writing.

