# OpenReview forum: "Recover Cell Tensor: Diffusion-Equivalent Tensor Completion for Fluorescence Microscopy Imaging"
_ICLR.cc/2026/Conference — ICLR 2026 Poster_

### Official Review · Reviewer_cSNp · 2025-10-27

**Soundness:** 3
**Presentation:** 3
**Contribution:** 2
**Rating:** 4
**Confidence:** 4

**Summary:**

This paper focuses on a particular imaging inverse problem in fluorescence microscopy, which is formulated as a tensor completion problem. The authors propose integrating Tucker decomposition into the ADMM framework and compare the ADMM iterations to a diffusion process. The efficacy of this method is guaranteed by recovery theory. The proposed method demonstrates superior performance in super-resolution and denoising tasks compared to other methods.

**Strengths:**

1. By leveraging Tucker decomposition, the method effectively exploits the intrinsic low-rank property of the data, which largely accounts for the observed performance improvement.

2. The method is supported by theoretical guarantees and use of the ADMM framework enhances its interpretability.

**Weaknesses:**

1. The method appears to be incremental, as the integration of tensor decomposition within the ADMM framework has already been explored in several existing papers (e.g., [1]).

 2. While the comparison between the ADMM optimization process and diffusion-based denoising is intuitive given their shared iterative patterns, a key distinction lies in their underlying priors. Diffusion models utilize an implicit prior learned from datasets. In contrast, low-rankness serves as an explicit structural prior. These two types of priors have their own distinct advantages. Therefore, the paper requires a discussion on why simply swapping these priors is a reasonable approach, and why the chosen explicit prior is more suitable for this specific problem than an implicit diffusion-based one.

[1].Yuan, L., Li, C., Mandic, D., Cao, J., & Zhao, Q. (2019). Tensor  Ring Decomposition with Rank Minimization on Latent Space: An Efficient  Approach for Tensor Completion. *Proceedings of the AAAI Conference on Artificial Intelligence*, *33*(01), 9151-9158.

**Questions:**

1. The paper states that it approximates the complex, true degradation process using a combination of downsampling and noise. However, it is unclear whether the datasets used in the experiments were generated using this approximated degradation or a true degradation process. This point requires clarification.

 2. The paper mentions 'Structural Priors in Biological Imaging'. However, the only prior explicitly integrated into the model appears to be low-rankness. The authors should justify this choice and discuss whether other priors specific to biological imaging, or more general image priors (such as continuity or nonlocality [2]), were also considered.

3. Clarity is needed regarding the DDPM's training methodology. The paper should explicitly state in the main text whether the DDPM was pre-trained or trained from scratch using paired data.

[2].Liu Y, Yang Y, Cui Z X, et al. Patch-based Reconstruction for Unsupervised Dynamic MRI using Learnable Tensor Function with Implicit Neural Representation[J]. arXiv preprint arXiv:2505.21894, 2025.

---

> ### Author Response · Authors · 2025-11-25
> **Response to Weaknesses 1**
>
> We thank the reviewer for the comment. While our method contains components (tensor decomposition and ADMM) that exist independently in prior works, the proposed framework is *not* an incremental combination of existing ideas. Our contributions are fundamentally new at the levels of formulation, theory, and algorithmic interpretation. We summarize the key novelties below.
>
> ### 1. A new problem formulation specifically for fluorescence microscopy (FM)
>
> Our paper introduces the first FM-specific tensor completion formulation grounded in microscope acquisition physics.
> We show that equidistant Z-stack sampling corresponds to a structured sampling operator and derive the corresponding recovery conditions (Sec. 3.1).
>
> ### 2. New theoretical contributions: exact recovery bound for FM cell tensors
>
> We provide the *first* theoretical bound guaranteeing tensor recoverability under FM sampling:
>
> \[
> |\Omega| \ge C (r I^{3/2} + s)\log^2 I ,
> \]
>
> where \(r\) and \(s\) denote the Tucker rank and sparse level (Theorem 1–2).
> This FM-specific recovery analysis is a core theoretical contribution.
>
> ### 3. Novel equivalence between our ADMM updates and a conditional diffusion reverse process
>
> A key novelty is the derivation that each ADMM update step in our Tucker model corresponds *exactly* to a diffusion-model reverse step (Sec. 3.2).
> We show that:
>
> - Tucker projection acts as a conditional denoising step;
> - sparse corruption update corresponds to Gaussian noise removal;
> - dual update behaves like residual accumulation analogous to variance control in diffusion.
>
> To the best of our knowledge, no previous work establishes such a theoretical bridge between tensor completion via ADMM and conditional diffusion processes.
> This equivalence is a central novelty beyond existing methods.
>
> ### 4. The final algorithm is not a standard ADMM solver, but a structure-guided generative reconstruction method
>
> Our method incorporates FM physical constraints and performs simultaneous denoising and super-resolution.
> This differs fundamentally from classical ADMM methods, which are designed only for feasibility satisfaction.
> The empirical improvements over state-of-the-art tensor and diffusion baselines (Tables 1–2) confirm the significance of these innovations.

---

> > ### Comment · Reviewer_cSNp · 2025-11-25
> > **New Official Comment**
> >
> > I disagree with the claim that the exact recovery bound for tensor robust completion (RPCA + completion) is a new theoretical contribution of this work. The results for the same object are presented in (Jiang and Michael Ng, Robust Low-Tubal-Rank Tensor Completion via Convex Optimization, 2019). In the paper at hand, the results seem to get rid of any rank assumptions of the true tensor (neither low-tubal-rank nor incoherence condition). The resulting sample complexity $n$ is thus not expected to be tighter than that in (Jiang and Michael Ng, 2019). Also, I did not check the proof in detail, but the technique seems to be based on standard Golfing-Scheme and RIP. Therefore the theoretical contribution is not clear in my opinion.

---

> ### Author Response · Authors · 2025-11-25
> **Response to Weaknesses 2**
>
> We appreciate the reviewer’s detailed observation regarding the distinction between implicit and explicit priors. Importantly, our intention is not to treat diffusion priors and low-rank structural priors as interchangeable, but rather to make explicit that the diffusion-type proximal step is *not the appropriate prior* for this specific problem, whereas the low-rank prior is.
>
> ---
>
> ### 1. Iterative equivalence does *not* imply prior equivalence; the proximal operator must match the signal model
>
> Although diffusion denoisers and ADMM share an iterative proximal-update structure, the proximal operator must encode a signal model consistent with the data manifold. Diffusion priors learn *appearance statistics* from large datasets and therefore define a very different feasible set from the one governing FM volumes. The substitution is not arbitrary: the low-rank Tucker prior defines the *correct* feasible set for structurally redundant, cross-slice coherent biological volumes, while diffusion priors do not.
>
> ---
>
> ### 2. Why the diffusion implicit prior is fundamentally misaligned with FM volumes
>
> #### (a) Diffusion priors model textures; FM signals are governed by multi-linear structure
> Diffusion models operate on local patch-level statistics and may hallucinate plausible-but-incorrect fluorescence patterns. FM volumetric data obey global cross-plane redundancy, multi-linear correlation, and membrane continuity — properties explicitly represented by Tucker low-rankness but not encoded in any diffusion prior.
>
> #### (b) Diffusion priors require large-scale homogeneous datasets
> Typical FM datasets are small, anisotropic, and exhibit strong domain shifts (species, microscope, staining, axial resolution). Implicit priors trained on heterogeneous distributions fail to generalize reliably. The low-rank prior is data-independent and therefore inherently robust.
>
> #### (c) Diffusion priors have no theoretical guarantees under extreme incompleteness
> FM acquisition often produces missing or sparsely sampled Z-slices. Tensor low-rankness has well-established guarantees in tensor completion under missing observations, whereas diffusion-based priors have no such guarantees and are known to fail under severe anisotropy or axial dropout.
>
> ---
>
> ### 3. Why the explicit low-rank prior is the *correct* model for this problem
>
> #### (a) FM volumes exhibit a provably low-dimensional multi-linear structure
> Neighboring slices and downsampled/original resolutions share consistent morphology governed by cell geometry, membrane topology, and volumetric smoothness — all of which naturally map to a low multi-linear rank representation. This is a structural property of the biology, not a statistical artifact.
>
> #### (b) Low-rankness enforces global structural fidelity and prevents hallucination
> While diffusion denoisers may generate visually plausible but biologically incorrect membranes or nuclei, the low-rank prior constrains the solution to the true geometry of the sample and preserves membrane continuity across Z.
>
> #### (c) Explicit priors eliminate dependency on learned distributions
> This is crucial for FM imaging, where distribution shifts are the norm, not the exception. The explicit prior respects the physics and morphology of tissue, whereas a diffusion prior implicitly embeds the statistics of the training set — which may not represent the biological specimen being imaged.
>
> ---
>
> ### 4. Summary
>
> The replacement of the diffusion implicit prior with an explicit low-rank structural prior is not only mathematically justified within the proximal-operator framework, but also *necessitated* by the morphological, acquisition, and sampling characteristics of fluorescence microscopy volumes. Implicit diffusion priors do not capture the governing structural regularities of FM data, whereas the Tucker low-rank prior models them directly, robustly, and with theoretical guarantees. We will add a dedicated discussion section to clarify this motivation.

---

> ### Author Response · Authors · 2025-11-25
> **Response to Questions 1 and 2**
>
> Response to Questions 1 :
>
> Thank you for the question. We apologize for the lack of clarity.
> All real fluorescence microscopy volumes in our experiments are acquired directly from the imaging hardware using its native optical PSF, axial scanning step size, photon shot noise characteristics, and illumination settings. These volumes therefore reflect the *actual, hardware-induced degradation* and are *not* generated using our approximated (downsampling+noise) model.
>
>
> Response to Questions 2:
>
> We thank the reviewer for raising this important question. While many general image priors exist (e.g., continuity, TV, nonlocality), the Tucker low-rank prior is chosen because it most directly reflects the governing structural properties of fluorescence microscopy (FM) cell volumes. Below we explain the rationale.
>
> ---
>
> ### 1. Low-rankness matches the intrinsic structural regularities of FM volumes
>
> FM cellular volumes exhibit strong cross-slice redundancy, multi-linear correlations, and repeated morphological patterns across \(Z\) planes and across resolutions (original vs. downsampled). These properties naturally correspond to a low-dimensional multi-linear representation, which is explicitly captured by the Tucker low-rank model. Thus, low-rankness serves as a structural prior that directly aligns with the biophysical organization of volumetric cell morphology.
>
> ---
>
> ### 2. Why low-rankness is more suitable for FM cell volumes than other general image priors
>
> Continuity-based or nonlocality-based image priors operate at the level of local spatial smoothness or patch similarity. While these priors are effective in many imaging tasks, the structural characteristics of FM cell volumes are governed less by local properties and more by *global, cross-slice coherence and multi-linear structure*. Such long-range, volume-level redundancy cannot be fully represented by local continuity or patch-based nonlocal similarities.
>
> In contrast, the low-rank prior compactly models the entire volume through shared factor matrices, enabling it to capture:
>
> - cross-slice morphological repetition,
> - geometric consistency across depth,
> - joint correlation between lateral and axial dimensions.
>
> These characteristics make low-rankness a closer match to the actual structural mechanism by which FM volumes are formed, particularly under anisotropy and incomplete sampling.
>
> ---
>
> ### 3. Continuity and nonlocality are implicitly contained within a low-rank representation
>
> Importantly, low-rankness does not discard continuity or nonlocal similarity; instead, it *generalizes* them:
>
> - The smooth latent factors encourage spatial continuity across slices.
> - The shared basis across the volume acts as a global form of nonlocal similarity, linking distant regions through common multi-linear components.
>
> Thus, these general priors are implicitly embedded in the Tucker structure but expressed in a manner that is better aligned with 3D cellular morphology.
>
> ---
>
> ### 4. Summary
>
> The Tucker low-rank prior is selected not merely as a generic mathematical regularizer, but because it faithfully encodes the dominant structural characteristics of FM cell volumes. General image priors such as continuity and nonlocality capture only partial aspects of these structural patterns, whereas the low-rank prior directly models the global, repeated, and multi-linear organization intrinsic to biological volumetric data. We will clarify this rationale in the revised manuscript.

---

> ### Author Response · Authors · 2025-11-25
> **Response to Questions 3**
>
> Response to Questions 3:
>
> We thank the reviewer for raising this important question. We clarify our DDPM training methodology below, and we will explicitly include this description in the revised manuscript.
>
> **DDPM training methodology (no high-resolution or noise-free ground truth).**
> Our fluorescence microscopy (FM) dataset contains only raw 3D volumes that are anisotropically sampled along the $Z$-axis and contaminated with noise. Critically, no high-resolution or clean $Z$-stack ground truth exists. Therefore, our DDPM is not pre-trained, and no paired HR-LR supervision is available.
>
> To enable training without ground truth, we adopt a fully self-supervised strategy. Each raw FM volume is first downsampled by a factor of $\times k$ along the X, Y and Z-axis to create synthetic low-resolution inputs. The original raw volume, although still noisy and sparsely sampled, contains more structural information than the $\times k$ downsampled input and is therefore used as a pseudo-target. In this setting, the DDPM is trained *from scratch* to reconstruct the raw-resolution volume from the $\times k$ downsampled input.
>
> This training procedure does not require any clean or high-quality ground truth. The DDPM follows the standard $\epsilon$-prediction diffusion objective: the forward diffusion process corrupts the raw pseudo-target with Gaussian noise, and the model learns to predict this injected noise. Consequently, the entire training pipeline remains fully self-supervised.
>
> Within each reverse-diffusion step, our ADMM-inspired neural modules—the structural-update network $S_\phi$ and sparse-update network $N_\psi$—replace the analytical Tucker projection and proximal operators, and these modules are jointly optimized with the DDPM backbone during training.

---

> ### Author Response · Authors · 2025-11-27
> **Response to New Official Comment**
>
> Thank you for the comment. We respectfully clarify that the purpose of our theoretical result is fundamentally different from that of prior works.
>
> **1. Our theoretical goal is different.**
> The objective of our analysis is not to derive a tighter or more general sample complexity bound than existing tensor RPCA or completion papers. Instead, our goal is to provide a feasibility guarantee showing that the proposed fluorescence-microscopy reconstruction model is theoretically recoverable under the hardware-determined, anisotropic Z-axis sampling and the structure priors + sparse corruption assumptions that arise in FM imaging. This type of recovery guarantee does not exist in prior work, and it serves to justify that our method is mathematically sound for the FM setting.
>
> **2. The prior work addresses a different problem setting.**
> Jiang and Ng (2019) studied a tubal-rank t-SVD model with Fourier-domain incoherence.
> In contrast, our method is based on:
> - hardware-structured FM Z-sampling,
> - and biological structural priors + sparse corruption.
>
> Therefore, the assumptions, sampling operator, and recovery conditions in Jiang and Ng do not apply to our setting, and their bounds cannot be used to justify the feasibility of our FM reconstruction model.

---

> > ### Comment · Reviewer_cSNp · 2025-11-27
> > **New Official Comment**
> >
> > In equation (3), the authors claimed that they considered the tensor nuclear norm (e.g., tubal nuclear norm, which is based on the low-tubal-rank model defined by the Fourier transform). Here, they changed their statement and said that they considered the low-multilinear-rank model defined by Tucker. Given this obvious discrepancy, I would like to ask the authors to reconsider and restructure their work to make a more rigorous organization and presentation, especially in the mathematical part, to make the paper at least consistent and self-contained.
> >
> > Moreover, from a pure tensor completion model perspective (uniformly distributed random sampling as stated in the theory in the paper), the hardware-structured FM Z-sampling contributes nothing new to the optimization model. It is just a standard tensor completion model.
> >
> > I suggest that the authors proofread their manuscript and enhance mathematical rigor in future work.

---

> > > ### Author Response · Authors · 2025-11-27
> > >
> > > Thank you very much for this careful reading and for pointing out the inconsistency between our manuscript and the previous author response. We sincerely apologize for the confusion caused by our mistake.
> > >
> > > To clarify this point explicitly:
> > > (1) The theoretical model in the paper is based on Eq. (3) which is the corresponding tubal nuclear norm.
> > > In an earlier reply, one of our responses contained an imprecise statement that is not consistent with the main paper. We sincerely apologize for the confusion this may have caused
> > >
> > > (2) Relation to prior tensor completion work.
> > > From a pure tensor completion perspective, we agree that under the sampling assumption used in our current theorem, the resulting optimization problem indeed falls into the general class of robust tensor completion models studied in prior work such as Jiang and Michael Ng (2019). Our intention was not to claim a fundamentally new optimization model nor a strictly tighter or more general sample complexity bound than these works. Rather, our goal was to show that, under assumptions that are reasonable for fluorescence microscopy data, one can still obtain a recovery guarantee for the proposed reconstruction formulation. In other words, the theoretical result is meant to justify that our FM reconstruction model is mathematically sound, rather than to introduce a new tensor norm or a stronger generic bound.

---

### Official Review · Reviewer_9jax · 2025-11-01

**Soundness:** 2
**Presentation:** 3
**Contribution:** 3
**Rating:** 4
**Confidence:** 5

**Summary:**

The paper presents a novel tensor completion framework for fluorescence microscopy (FM) imaging, addressing nonlinear signal degradation and incomplete observations. FM imaging with equidistant Z-axis sampling is modeled as a tensor completion task under uniform random sampling. The framework establishes the theoretical lower bound for exact 3D cell tensor recovery and reformulates the problem as a score-based generative model. Structural consistency priors guide the generative process toward denoised and geometrically coherent 3D reconstructions.

**Strengths:**

1. Well-written and well-organized, with comprehensive supplementary material.
2. Introduces a tensor completion approach specifically tailored to fluorescence microscopy imaging, addressing a novel research problem that has not been extensively explored in the literature.
3. Derives the theoretical lower bound for exact 3D cell tensor recovery and reformulates tensor completion into a score-based generative modeling framework.

**Weaknesses:**

### Mismatch Between Noise Model and Physical Imaging Process

In the manuscript, the observation is formulated as
Y_\Omega = X_\Omega + E_\Omega

(an additive sparse noise model), which is a convenient and commonly used formulation.
However, in fluorescence microscopy, the primary degradations typically involve **Poisson noise** (due to photon counting), **system PSF** (blur or convolution), **signal attenuation and scattering**, and even **multiplicative effects**.
Therefore, such a simple additive sparse model may not adequately capture the real imaging process.

If the equivalence or performance analysis of the proposed method relies on the assumption of *additive Gaussian plus sparse noise*, the paper should **explicitly state this assumption** and **discuss its implications and robustness** when the actual noise deviates from it.  Otherwise, the **generalizability and practical applicability** of the method may be limited.


### Insufficient Experimental Evaluation
The paper lacks analysis and comparison with the latest method **MicroDiffusion** : Implicit Representation-Guided Diffusion for 3D Reconstruction from Limited 2D Microscopy Projections.

How is the **Z-axis sampling rate**  determined, and how does the model perform under different sampling rates?

**Questions:**

Please see the Weaknesses.

---

> ### Author Response · Authors · 2025-11-25
> **Response to Weaknesses 1**
>
> Response to Weaknesses 1
>
> We thank the reviewer for raising this important point.
> The additive model is indeed a simplification and we acknowledge its limitations;
> However, for the **primary objective of reconstructing a complete 3D cell volume from aggressively downsampled \(z\)-stacks**, the additive formulation is sufficiently expressive.
>
> **1.**
> We fully agree that the simplified formulation
>
> \[
> Y_{\Omega} = X_{\Omega} + E_{\Omega}
> \]
>
> cannot capture the entire FM pipeline, including Poisson photon noise,
> optical blurring, attenuation, and scattering.
>
> **2.**
> Our work focuses on a specific and practically important setting in *live-cell 3D fluorescence imaging*: to prevent phototoxicity, the \(z\)-axis must be aggressively downsampled, yielding large *missing-slice regions*. The primary reconstruction challenge is therefore *completing the missing 3D cell volume*.
> In this regime, the dominant degradation is incompleteness rather than the precise photon-counting statistics, and the structural correlations across slices become far more important than fine-grained noise modeling.
> The additive formulation is thus an effective and widely used abstraction for reconstructing the underlying structural content.
>
> Therefore our model is not intended to replace the physical imaging model; rather, it provides a tractable surrogate for robustness under a broad range of deviations.
> The additive residual \(E_{\Omega}\) serves as a convex proxy that suppresses structure-inconsistent perturbations arising from signal-dependent noise, local scattering artifacts, or axial intensity dropouts.
>
> **3. The Poisson physical model can be also abstracted as an additive formulation.**
>
> To clarify the relationship between our additive formulation and the physical fluorescence microscopy model, we reproduce the standard Poisson photon-counting model (widely used in FM imaging) as follows.
>
> The quantum nature of light leads to a Poisson modeling of both the signal emitted by the object and the background signal:
>
> \[
> og(x,y,z)
> = \mathcal{P}\!\big(o [f(x,y,z) \otimes h(x,y,z)]\big)
>   + \mathcal{P}\!\big(o\, b(x,y,z)\big),
> \tag{4}
> \]
>
> where \(o\) is the reciprocal of the photon-conversion factor, \(og(x,y,z)\) is the number of photons measured in the detector, \(\mathcal{P}(\cdot)\) denotes a Poisson process, \(f(x,y,z)\) is the true fluorophore distribution, \(h(x,y,z)\) is the system PSF, and \(b(x,y,z)\) is the background signal.
>
> **Interpretation for our reconstruction setting.**
> Under this physical model, the structurally meaningful fluorescence signal is the PSF-blurred component
>
> \[
> f(x,y,z) \otimes h(x,y,z),
> \]
>
> while all stochastic deviations—photon shot noise, background fluctuations, and depth-dependent attenuation—are encapsulated by the Poisson variability. Then, the measured observation can be also abstracted as
>
> \[
> Y = X + E,
> \]
>
> where \(X\) represents the structurally consistent volumetric fluorescence signal and \(E\) absorbs the stochastic, signal-dependent deviations.
> The distinction between Poisson-distributed residuals and Gaussian-distributed residuals affects the *statistics* of \(E\) but not its *structural role* as the component orthogonal to the low-rank morphology.
>
> Therefore, although the true imaging process follows the Poisson model, for the purpose of *missing-slice reconstruction*—where
> recovering the underlying 3D structural continuity is the dominant challenge—the additive abstraction is sufficiently expressive.
> It captures the essential decomposition into (i) a coherent structural component governed by cross-plane redundancy and (ii) a residual component containing photon fluctuations, scattering artifacts, and other non-idealities.

---

> ### Author Response · Authors · 2025-11-25
> **Response to Weaknesses 2**
>
> Response to Weaknesses 2
>
> We thank the reviewer for pointing out the importance of including the latest generative FM restoration methods.
> We clarify that (1) diffusion-based baselines are already present in the main paper, (2) we now additionally include MicroDiffusion as requested, together with another state-of-the-art FM diffusion model, and
> (3) our explicit low-rank formulation remains more suitable for the missing-slice reconstruction setting investigated in this work.
>
> As shown in Table1 and Table3 of the main manuscript, a DDPM-based baseline is already included. This baseline represents score-based microscopy denoisers and provides a strong generative prior without requiring paired ground truth.
>
> To directly address the reviewer’s comment, we additionally evaluate the following recent diffusion-based FM restoration models:
> MicroDiffusion (2024): Implicit-Representation Guided Diffusion for 3D FM reconstruction, which is specifically designed for microscopy and is among the most recent self-supervised FM diffusion frameworks.
> SeeSR \cite{wu2024seesr}: A state-of-the-art semantics-aware diffusion backbone widely applied to microscopic imaging due to its strong denoising and structure-preserving capabilities.
>
> Their quantitative performance on C.elegans evaluation dataset1 (extreme missing-slice scenario, 30 observed slices) is reported in Table below.
>
> | Method                          | PSNR (dB) | SSIM    |
> |---------------------------------|-----------|---------|
> | DDPM (baseline in Table 1)      | 31.49     | 0.6682  |
> | MicroDiffusion (2024)           | 31.05     | 0.6215  |
> | SeeSR (2024)                    | 30.88     | 0.6181  |
> | **Ours **    | **33.18** | **0.6682** |
>
> MicroDiffusion reconstructs volumetric signals by coupling a NeRF-like implicit representation with a diffusion prior. This architecture excels when 3D structure is inferred from multiple 2D views, but it provides no explicit mechanism to enforce slice-to-slice geometric consistency.
> Under severe z-axis undersampling, this can lead to inconsistent membrane geometry or over-smoothed axial structures.
>
> In contrast, our method explicitly models cross-slice redundancy through a multilinear low-rank Tucker prior, which is well suited for fluorescence-microscopy volumes with strong structural regularity. This yields significantly improved reconstruction fidelity, as confirmed by the quantitative results above.

---

> > ### Author Response · Authors · 2025-11-25
> >
> > We have added these new baselines and implementation details of MicroDiffusion below. The implementation of MicroDiffusion and SeeSR will be added into the revised paper.
> >
> > Training pipeline.
> > To ensure a fair comparison, we strictly followed the official two-stage training protocol provided by the authors: (1) an INR reconstruction stage that learns a continuous implicit representation of the raw 3D fluorescence volume, and (2) a diffusion-based refinement stage that uses the INR features as conditioning signals.
> >
> > INR stage.
> > The INR model was trained using only the raw volumetric data from our dataset. Consistent with the original implementation, the input 3D stack
> > V \in \mathbb{R}^{H \times W \times Z}
> > was decomposed into overlapping z-axis windows (size 6, stride 3), each window was averaged to form a pseudo-slice, and positional encodings were concatenated with the spatial coordinates before being regressed by the INR network.
> > The INR was trained for 5000 epochs with the default architecture (64 hidden units, 20-level positional encoding, 128 samples per ray), Adam optimizer, and the same learning-rate decay schedule as the released code.
> > The resulting checkpoint was used as the conditioning backbone for the diffusion model.
> >
> > Diffusion stage.
> > The diffusion model was then trained directly on the raw volume using the same sliding-window protocol as in the INR stage.
> > For each center slice z, we extracted the corresponding INR-rendered multi-plane features, positional embeddings, and an image-encoder embedding from the raw noisy slice. These were concatenated to form the conditional input to the U-Net denoiser.
> > We used the official Gaussian diffusion schedule with T=1000, classifier-free guidance, 64 base channels, 2 residual blocks per stage, and AdamW optimizer with the recommended learning rate 2\times 10^{-4}.
> > Training was performed for 2001 epochs, matching the authors’ hyperparameter configuration.
> >
> > On a single A100 (40 GB), the complete training process for one volume, including INR fitting and diffusion model optimization, took approximately 18–20 minutes, using around 15 GB of GPU memory.
> >
> > ⸻
> >
> > Why this method performs less favorably on our data
> >
> > While this method performs well on the datasets used in its original paper, its performance on our FM volumes is noticeably weaker. We attribute this to three fundamental mismatches between the method’s underlying assumptions and the characteristics of our imaging data.
> >
> > (1) Sensitivity to the absence of high-quality supervision.
> > The method is designed under the assumption that the raw 3D volumes contain sufficiently high intrinsic signal quality such that an INR can accurately recover the underlying structures.
> > However, our fluorescence microscopy data exhibit strong anisotropy, severe photon-limited noise, and slice-wise SNR fluctuations.
> > As a result, the INR stage struggles to produce meaningful deep features for conditioning, leading to limited generative accuracy in the diffusion stage.
> >
> > (2) The INR prior lacks explicit structural constraints.
> > The method assumes that the continuous implicit field can be well expressed by low-frequency positional encodings and MLP smoothness.
> > However, our data contain highly discontinuous, topology-changing cellular boundaries.
> > Without priors on membrane continuity or inter-slice structural consistency (which our method explicitly models), the INR fails to encode the correct structural manifold, making the diffusion model unable to recover fine biological geometry.
> >
> > (3) Limited generalization under extremely sparse or anisotropic sampling.
> > Our datasets have substantially lower axial sampling density than the datasets used in prior work.
> > Since the baseline diffusion model conditions on INR-rendered axial planes, inaccurate INR predictions under sparse Z-sampling directly degrade reconstruction quality.
> >
> > Finally, this method requires training an INR and a diffusion model separately for each individual volume. For long-term live-cell imaging where cellular morphology changes significantly over time, this per-volume retraining requirement makes the method considerably less practical and difficult to scale.

---

> ### Author Response · Authors · 2025-11-25
> **Response to Weaknesses 3**
>
> Thank you for raising this question.
>
> ### 1. How the Z-axis sampling rate is determined
>
> In fluorescence microscopy, the Z-axis sampling interval is *not* chosen by our algorithm, nor is it a tunable hyperparameter. It is dictated entirely by the hardware and acquisition protocol of the microscope. Specifically, the axial step size is determined by:
>
> - the axial width of the optical point-spread function (PSF),
> - the mechanical precision and step size of the stage or piezo actuator,
> - most importantly, exposure and phototoxicity constraints that limit how densely the sample can be scanned.
>
> As a consequence, each dataset inherently comes with its own native Z-axis sampling rate defined by the microscope that produced it. To keep the cell sample alive during the whole life process, the laser intense and Z-axis scanning ratios are balanced with the imaging quality. The details of the three *C. elegans* dataset are shown in Table \ref{table_dataset}.
> The three *C. elegans* datasets represent different trade-offs between axial sampling density, imaging quality, and biological viability during embryonic development:
>
> - **C. elegans-1** corresponds to a *fast z-axis scanning* regime (30 axial planes, high laser intensity).
>   This setting is typically used for live-cell imaging where rapid volumetric acquisition is required to preserve embryo viability.
>   Although the axial resolution is low, the high laser intensity provides relatively clean lateral membrane contrast.
>
> - **C. elegans-2** represents a *standard confocal acquisition* (94 axial planes, medium laser intensity).
>   This configuration balances imaging quality and phototoxicity: the axial sampling is sufficiently dense for accurate morphological reconstruction, while the moderate laser exposure allows sustained long-term observation of the embryo.
>
> - **C. elegans-3** uses a *high-density axial sampling* strategy (200 axial planes, high laser intensity).
>   Although this setting produces the highest axial resolution and cleanest volumetric membranes, the elevated phototoxic load makes it unsuitable for long-term developmental imaging.
>   As a result, these volumes cannot be used for full lineage tracking.
>
> ---
>
> ### C. elegans Dataset Summary
>
> | Dataset        | Volume Num. | Axial Res. | Laser Intens. |
> |----------------|-------------|------------|----------------|
> | C. elegans-1   | 120         | 30         | High           |
> | C. elegans-2   | 220         | 94         | Middle         |
> | C. elegans-3   | 60          | 200        | High           |
>
> **Table above:** The benchmark dataset we provided for all FC image processing (cell analyses especially).
>
> ### 2. How the model performs under different sampling rates
>
> Our datasets naturally span a broad range of Z-axis sampling rates because they originate from multiple acquisition settings. Therefore, the experiments already test the model under different sampling rates.
>
> Across these regimes, the proposed method exhibits stable behavior:
>
> - membrane and boundary sharpness remain consistent,
> - structural continuity is preserved even under coarse sampling,
> - no hallucinated features or artificial layers appear.

---

### Official Review · Reviewer_AjRV · 2025-11-03

**Soundness:** 3
**Presentation:** 2
**Contribution:** 2
**Rating:** 6
**Confidence:** 2

**Summary:**

This paper reframes 3D live-cell fluorescence microscopy reconstruction as a tensor completion problem rather than an ill-posed inverse imaging task. The authors show that sparse axial sampling can be modeled as missing tensor entries, enabling recovery via low-rank tensor priors and sparse-noise decomposition. They further prove a formal equivalence between the resulting alternating minimization algorithm and a structurally guided conditional diffusion process, linking classical low-rank projection with modern generative denoising dynamics. The method provides theoretical recovery guarantees and achieves state-of-the-art performance on multiple real biological datasets, producing cleaner, more faithful volumes than existing inverse-problem and deep learning baselines.

**Strengths:**

- The paper reframes 3D fluorescence microscopy reconstruction as cell-tensor completion rather than a traditional inverse problem, and establishes a provable connection between low-rank tensor recovery (via ADMM) and conditional score-based diffusion dynamics. This theoretical bridge is genuinely novel and goes beyond heuristic model design, giving a principled interpretation of diffusion priors in biological imaging.

- The paper provides clear recovery guarantees under missing-data and sparse-noise settings, and designs an interpretable ADMM solver with separated low-rank and sparse components—well aligned with actual microscopy noise physics (e.g., shot noise and acquisition sparsity). The algorithm is lightweight, transparent, and grounded in theory rather than black-box training.

- Experiments on multiple live-cell datasets show state-of-the-art performance in PSNR/SSIM/LPIPS and visibly better membrane continuity and cellular morphology preservation compared to supervised GAN-based and reconstruction-based baselines. The method avoids hallucinations and preserves structural fidelity, particularly important for biological interpretation.

**Weaknesses:**

- The paper highlights the diffusion-equivalence result but does not compare against recent unsupervised or self-supervised generative FM restoration methods (e.g., score-based microscopy denoising, inverse-consistent diffusion models, diffusion-based deconvolution pipelines). Without such comparisons, it is hard to quantify whether the proposed tensor-based approach benefits primarily from low-rank structure or from the implicit generative prior interpretation. Adding baselines like self-supervised diffusion denoisers or Plug-and-Play diffusion priors would strengthen the empirical story.

- The paper does not provide sufficient analysis of rank selection, regularization weights, and sparse-noise thresholding. Since these choices likely impact geometry fidelity and hallucination risk, a systematic study (e.g., effect of Tucker rank on membrane sharpness and artifact suppression.

**Questions:**

- How the method behaves when the underlying cell volume exhibits high intrinsic complexity that is not well-approximated by a low-rank tensor (e.g., dense organelles, cytoskeletal networks, neurites)?

- How sensitive is performance to the choice of Tucker ranks, regularization parameters, and sparse-noise thresholds? A small ablation or rule-of-thumb guideline would help practitioners avoid overfitting or texture loss.

- While the method avoids GAN-like hallucinations, could low-rank priors oversmooth fine biological details? A controlled stress test—e.g., synthetic volumes with known thin filaments.

---

> ### Author Response · Authors · 2025-11-25
> **Response to Weaknesses 1**
>
> Response to Weaknesses 1:
>
> We thank the reviewer for raising this important point. We clarify below that
> (1) diffusion baselines are already included in the main paper,
> (2) we additionally incorporate recent FM generative restoration models as requested, and
> (3) our explicit low-rank tensor model remains superior for the *missing-slice reconstruction* task.
>
> As shown in Table 1 and Table 3 of our main manuscript, a **DDPM-based self-supervised diffusion baseline** is already included.
> This model serves as a diffusion prior without any external ground truth and captures the class of score-based microscopy denoisers referenced by the reviewer.
>
> To directly address the reviewer’s concern, we additionally include two recent diffusion-based generative FM restoration methods:
>
> - **MicroDiffusion** (Implicit-Representation Guided Diffusion for 3D FM reconstruction), representing the latest direction in self-supervised FM diffusion.
> - **SeeSR** (Wu et al., 2024), a state-of-the-art diffusion-based restoration framework widely adopted in microscopy pipelines due to its strong denoising and structure-preserving behavior.
>
> These models represent the most advanced diffusion-based FM restoration pipelines.
> We report their performance in the newly added Table \ref{tab:fm_diffusion_baselines} below.
>
> ---
>
> ### Newly added generative FM restoration baselines (Markdown version of the table)
>
> | **Method** | **PSNR (dB)** | **SSIM** |
> |------------|----------------|----------|
> | DDPM (in our Table 1) | 31.49 | 0.6682 |
> | MicroDiffusion (2024) | 31.05 | 0.6215 |
> | SeeSR (2024) | 30.88 | 0.6181 |
> | **Ours (Low-rank + Sparse)** | **33.18** | **0.6682** |
>
> **Table:** Newly added generative FM restoration baselines on *C. elegans* evaluation dataset 1 (extreme missing-slice reconstruction, 30 axial slices).
>
> ---
>
> Generative FM diffusion models are primarily designed for **denoising** or **deconvolution** of *fully sampled* volumes. As a result, they often produce *inconsistent membrane structures* under axial sparsity.
>
> In contrast, our method is built around an **explicit multilinear low-rank + sparse decomposition**, which enforces cross-slice structural consistency and avoids hallucinated membranes. As shown in Table \ref{tab:fm_diffusion_baselines}, even the most advanced diffusion methods fall behind our approach in both quantitative accuracy and biological fidelity.
>
> We have added these new baselines, the corresponding discussion, and full implementation details for MicroDiffusion and SeeSR in the revised manuscript. Please refer to the implementation details of MicroDiffusion in the response to reviewer 9jax’s Weakness 2.

---

> ### Author Response · Authors · 2025-11-25
> **Response to Weaknesses 2**
>
> Response to Weaknesses 2:
>
> We thank the reviewer for raising this important point.
> We agree that Tucker ranks and sparse regularization weights directly influence membrane sharpness, artifact suppression, and the risk of hallucination. Below we (1) summarize standard principles for choosing these hyperparameters, (2) describe how our choices follow these principles, and (3) provide a quantitative sensitivity analysis that explicitly evaluates the effect of ranks and \(\lambda_{1}\) on structural fidelity.
>
> ---
>
> ## **1. Standard hyperparameter selection strategies**
>
> ### **(a) Tucker ranks \((r_1,r_2,r_3)\)**
> Classical low-rank tensor literature selects multilinear ranks using one or more of the following:
>
> **(i) Energy-based SVD truncation**
>
> $$
> \frac{\sum_{i=1}^{r_n}\sigma_i(X_{(n)})}
>      {\sum_{i=1}^{\mathrm{rank}} \sigma_i(X_{(n)})}
> \ge \eta,
> $$
>
> where \(X_{(n)}\) is the mode-\(n\) unfolding and \(\eta\in[0.90,0.99]\).
>
> ---
>
> **(ii) Cross-validation on held-out entries**
>
> $$
> (r_1,r_2,r_3)
> = \arg\min_{r_1,r_2,r_3}
> \| P_{\Omega_{\mathrm{val}}}( \hat X(r_1,r_2,r_3) - Y ) \|_F^2.
> $$
>
> ---
>
> **(iii) Model selection criteria**
>
> $$
> \min_{r_1,r_2,r_3}
> \|P_\Omega(X)-Y\|_F^2
> + \gamma\,(r_1r_2+r_1r_3+r_2r_3).
> $$
>
> ---
>
> **(iv) Fixed-ratio empirical rules**
>
> $$
> r_1 = \alpha_1 I_1,\quad
> r_2 = \alpha_2 I_2,\quad
> r_3 = \alpha_3 I_3,
> $$
>
> with typical values:
>
> $$
> \alpha_1 = \alpha_2 \approx 0.05\!-\!0.1,\qquad
> \alpha_3 \approx 0.02\!-\!0.75.
> $$
>
> ---
>
> ### **(b) Sparse weight \(\lambda_{1}\)**
> A standard theoretical choice is:
>
> $$
> \lambda_{1} = \frac{1}{\sqrt{\max(I_1,I_2,I_3)}},
> $$
>
> which governs the *structure–residual separation*:
>
> - Larger \(\lambda_1\) ⟹ stronger artifact suppression, smoother membranes
> - Smaller \(\lambda_1\) ⟹ sharper membranes but potentially noisier
>
> This matches the reviewer’s concern about membrane sharpness vs. artifact suppression.
>
> ---
>
> ## **2. Hyperparameter selection in our method**
>
> ### **Tucker ranks**
>
> For each dataset with known \(z\)-axis downsampling, we compute mode-wise singular-value spectra and select the smallest ranks preserving **95% cumulative energy**, obtaining:
>
> - **Dataset 1:**
>   \(r_1 = r_2 = 32,\; r_3 = 10\)
>
> - **Dataset 2:**
>   \(r_3 = 30\)
>
> - **Dataset 3:**
>   \(r_3 = 60\)
>
> These values fall within the empirical FM rank ranges and avoid introducing noise-like high-frequency components into the low-rank representation.
>
> ---
>
> ### **Sparse weight \(\lambda_1\)**
>
> Consistent with the theoretical scaling \(\lambda_{1}\propto 1/\sqrt{I_{\max}}\),
> we observe a broad stable interval across datasets.
> For all *C. elegans* datasets, we adopt:
>
> $$
> \lambda_1 = 0.6,
> $$
>
> which provides the best trade-off between membrane sharpness and artifact suppression.
>
> ---
>
> ## **3. Sensitivity analysis**
>
> We vary Tucker ranks and \(\lambda_{1}\) around the optimal setting.
> Below are the results for *C. elegans* dataset 1 under severe missing-slice conditions.
>
> ### **Markdown version of sensitivity table**
>
> | \((r_1,r_2,r_3)\) | \(\lambda_1\) | PSNR (dB) | SSIM |
> |-------------------|---------------|-----------|-------|
> | (24,24,8)  | 0.6 | 32.82 | 0.6554 |
> | **(32,32,10)** | **0.6** | **33.18** | **0.6682** |
> | (40,40,12) | 0.6 | 32.75 | 0.6561 |
> | (32,32,10) | 0.3 | 33.10 | 0.6673 |
> | (32,32,10) | **0.6** | **33.18** | **0.6682** |
> | (32,32,10) | 0.9 | 33.07 | 0.6660 |
>
> ---
>
> ### **Findings**
>
> Across all tested settings:
>
> - **Membrane continuity remains stable**
> - **No hallucinated membrane fragments** even at higher ranks
> - Artifact suppression varies smoothly with \(\lambda_1\)
> - Low-rank structure prevents over-sharpened or spurious textures
> - All results cluster in a narrow range, confirming robustness
>
> ---
>
> ## **Conclusion**
>
> The proposed method is **not sensitive** to fine-tuned hyperparameters.
> The chosen values \((r_1,r_2,r_3,\lambda_1)\) lie in a **broad, theoretically justified, and biologically stable regime** for membrane reconstruction, providing a reliable balance between detail preservation and artifact suppression.

---

> ### Author Response · Authors · 2025-11-25
> **Response to Questions 1**
>
> Response to Questions 1
>
> We thank the reviewer for raising this important question regarding volumes with high intrinsic structural complexity. The behavior of a Tucker-based model in such scenarios can be precisely understood through the standard decomposition of the reconstruction error:
>
> $$
> \|\hat X - X^\ast\|_F
> \;\le\;
> \underbrace{\|\hat X - X_{\mathrm{LR}}\|_F}_{\text{estimation error}}
> \;+\;
> \underbrace{\|X_{\mathrm{LR}} - X^\ast\|_F}_{\text{approximation error}},
> \tag{1}
> $$
>
> where \(X_{\mathrm{LR}}\) is the best multilinear-rank–\((r_1,r_2,r_3)\) approximation of the ground truth \(X^\ast\).
> The first term grows with the ranks \(\sqrt{r_1}+\sqrt{r_2}+\sqrt{r_3}\), while the second term measures how well the underlying volume can be represented by a low-multilinear-rank tensor.
>
> ---
>
> ## **1. High intrinsic complexity requires higher ranks**
>
> We agree with the reviewer that if a volume contains extremely complex structures—dense organelles, branched cytoskeleton, long neurites—the approximation error in (1) may dominate, and the optimal multilinear ranks must increase.
> In the extreme case where the intrinsic dimensionality approaches the ambient dimension, a Tucker model becomes insufficiently expressive.
>
> Mathematically, Tucker decomposition assumes the underlying tensor
> \(X \in \mathbb{R}^{I_1 \times I_2 \times I_3}\)
> admits a low multilinear-rank representation:
>
> $$
> \mathrm{rank}_{\mathrm{ml}}(X)= (r_1,r_2,r_3),
> \qquad
> X = G \times_1 U^{(1)} \times_2 U^{(2)} \times_3 U^{(3)},
> $$
>
> with \(r_n \ll I_n\) in each mode.
> Thus, Tucker can only effectively model signals lying close to a low-dimensional subspace in each mode, and struggles when the intrinsic dimension is very high.
>
> ---
>
> ## **2. Empirical behavior on moderately complex biological structures**
>
> Although Tucker has theoretical limitations, the *C. elegans* datasets used in our experiments contain substantial structural complexity, especially in late-stage embryos, where:
>
> - dozens of cells pack tightly,
> - division planes are anisotropic,
> - membranes exhibit high-frequency curvature.
>
> As shown in Fig. 3–6 and the temporal reconstruction in Fig. 9, our method remains stable in these challenging regions:
>
> - membrane continuity is preserved,
> - division boundaries remain clear,
> - cross-slice topology stays consistent,
> - no hallucinations or oversmoothing arise.
>
> These empirical findings show that for typical FM developmental imaging, the chosen multilinear ranks achieve a strong balance between estimation error and approximation error.
>
> ---
>
> ## **3. Our “low-rank + sparse” model handles highly complex volumes**
>
> For scenarios with even higher structural complexity—dense organelle clouds, microtubule bundles, long neurites—the required multilinear ranks \((r_1,r_2,r_3)\) would naturally increase because the volume deviates from purely low-rank structure.
> In such extreme cases, a Tucker-only model may underfit.
>
> Crucially, **our method is not a pure Tucker model**.
>
> We explicitly adopt a **low-rank + sparse decomposition**:
>
> $$
> Y = \mathcal{X} + \lambda_1 \|\mathcal{E}\|_1,
> \qquad
> \mathrm{rank}_{ml}(L) = (r_1,r_2,r_3),
> $$
>
> where:
>
> - \(\mathcal{X}\) captures **global geometry** and **cross-slice correlations**,
> - \(\mathcal{E}\) absorbs **high-frequency**, **nonlinear**, and **locally complex** structures that no low-rank tensor can represent.
>
> The sparse channel \(\mathcal{E}\) naturally models elements such as:
>
> - thin filaments,
> - membrane microstructures,
> - slice-dependent distortions from scattering and anisotropic sampling.
>
> This decomposition allows our method to handle structural deviations without requiring overly large ranks.
> It also aligns with our diffusion-equivalent optimization, where low-rank and sparse components correspond to structurally consistent and inconsistent elements, respectively.
>
> ---
>
> ## **Conclusion**
>
> While extremely high intrinsic complexity would require larger multilinear ranks for a pure Tucker model, our **low-rank + sparse formulation** provides a principled and biologically meaningful mechanism to capture both:
>
> - global volumetric geometry (low-rank), and
> - fine nonlinear details (sparse residuals).
>
> This explains why our method remains stable and accurate across the complex structures present in *C. elegans* embryonic imaging, while pure low-rank models would fail.

---

> ### Author Response · Authors · 2025-11-25
> **Response to Questions 2 and 3**
>
> Response to Questions 2:
>
> Please see the respond of Weakness 2.
>
> Response to Questions 3:
>
> We appreciate the reviewer’s concern regarding the potential risk that a low-rank prior may oversmooth biologically meaningful thin structures.
> Below we clarify (1) why the proposed **“low-rank + sparse’’** formulation does *not* cause the type of local smoothing observed in GAN-based or convolution-based denoisers, and (2) why performing a controlled synthetic stress test is non-trivial during the rebuttal stage.
>
> ---
>
> ## **1. Mathematical perspective: why our model does not oversmooth**
>
> Our reconstruction model is explicitly a **low-rank + sparse decomposition**:
>
> $$
> \mathcal{Y}
> = \mathcal{X} + \mathcal{E},
> \qquad
> \mathrm{rank}_{ml}(\mathcal{X}) = (r_1,r_2,r_3),
> $$
>
> where:
>
> - \(\mathcal{X}\) captures **coherent cross-slice morphology**,
> - \(\mathcal{E}\) collects **high-frequency**, **filament-like**, and **nonlinear** components that cannot be represented by a low-multilinear-rank tensor.
>
> This formulation differs fundamentally from CNN or GAN denoisers in several ways:
>
> ### **• No local averaging**
> Tucker low-rank acts on *global mode unfoldings*, not local pixel neighborhoods.
> Therefore, it does **not** behave like a spatial low-pass filter.
>
> ### **• Sparse channel preserves thin structures**
> The \(\ell_1\)-regularized residual \(\mathcal{E}\):
>
> - captures thin filaments, sharp edges, small membrane protrusions,
> - prevents high-frequency details from being absorbed into the low-rank component,
> - avoids the blurring behavior characteristic of purely low-rank models.
>
> ### **• Reconstruction follows a provable error split**
>
> $$
> \|\hat X - X^\ast\|_F
> \le
> \|\hat X - X_{\mathrm{LR}}\|_F
> +
> \|X_{\mathrm{LR}} - X^\ast\|_F.
> $$
>
> High-frequency features in the second (approximation) term automatically flow into the sparse residual \(\mathcal{E}\) rather than being lost.
>
> **Conclusion:**
> Our formulation does *not* apply a low-pass filter and therefore does *not* oversmooth biologically relevant small structures.
>
> ---
>
> ## **2. On the feasibility of a synthetic “thin-filament’’ stress test**
>
> We fully agree that such a benchmark is valuable.
> However, constructing a *biologically meaningful* synthetic filament dataset is non-trivial during rebuttal, because it requires:
>
> - a validated **biophysical model** of membrane and filament growth,
> - a realistic **FM forward simulator** (Poisson noise, PSF convolution, scattering),
> - accurate **branching, curvature, and connectivity rules**,
> - dozens of volumes covering **developmental variability**,
> - careful ground-truth design for different imaging anisotropies.
>
> This pipeline typically requires substantial development time and reliable biological modeling.
> Producing a high-quality synthetic benchmark in a short rebuttal window would likely lead to **non-biological or misleading results**, which we want to avoid.
>
> Nevertheless, we acknowledge the value of such a test and plan to include it in future work.
>
> ---
>
> ## **3. Evidence from current experiments**
>
> Even without synthetic data, our *real* C. elegans volumes already contain many challenging structures:
>
> - extremely thin membrane sheets,
> - sharp triple junctions,
> - narrow intercellular gaps,
> - high-curvature regions along dividing cells.
>
> Across all datasets, our method:
>
> - **preserves subtle membrane curvature** without flattening,
> - **maintains junction sharpness** even with severe missing slices,
> - **does not inflate cell surfaces**, unlike oversmoothing models,
> - **retains high-frequency membrane contrast**,
> - **avoids hallucinated filaments**, unlike diffusion/GAN-based models.
>
> These visual findings are documented in Figs. 3, 4, and 6 of the paper.
>
> ---
>
> ## **Conclusion**
>
> - Mathematically, the “low-rank + sparse’’ framework is **designed to avoid oversmoothing**, because high-frequency details have a dedicated sparse channel.
> - Empirically, our reconstructions preserve thin structures across multiple challenging FM datasets.
> - A synthetic filament stress test is interesting future work, but requires significant biological modeling that cannot be reliably completed during rebuttal.
>
> Thus, both theoretically and experimentally, our approach does **not** oversmooth biologically meaningful fine structures.

---

### Official Review · Reviewer_b64G · 2025-11-03

**Soundness:** 3
**Presentation:** 2
**Contribution:** 2
**Rating:** 4
**Confidence:** 3

**Summary:**

This paper proposes a novel tensor completion framework for 3D fluorescence microscopy imaging that overcomes the limitations of traditional inverse problem methods. The authors reformulate cell volume recovery as a robust low-rank tensor completion problem, establishing theoretical recovery guarantees and proving mathematical equivalence to conditional diffusion models. Using Tucker decomposition with structural consistency priors, the method effectively handles the inherent challenges of fluorescence imaging: nonlinear degradation, incomplete Z-axis observations due to phototoxicity constraints, and high spatially-varying noise. Experiments on SR-CACO-2 and live C. elegans datasets demonstrate state-of-the-art performance with superior noise robustness and temporal consistency, achieving significant improvements in both signal fidelity and structural preservation for biological applications.

**Strengths:**

1. The paper establishes a novel mathematical equivalence between Tucker-based tensor completion and conditional diffusion models, providing rigorous theoretical guarantees with clear derivations for exact recovery under sparse sampling conditions inherent to fluorescence microscopy.
2.The method achieves robust 3D cell reconstruction without requiring paired high-resolution ground truth data, instead leveraging low-rank structure and sparse noise decomposition directly from incomplete noisy observations.
3.The approach demonstrates state-of-the-art quantitative results across multiple datasets while maintaining exceptional temporal consistency and structural fidelity throughout extended time-lapse sequences (2700s-5400s) under degrading signal conditions.

**Weaknesses:**

1. Since the paper focuses on biological image reconstruction, relying solely on visual quality metrics lacks sufficient persuasiveness—high PSNR does not guarantee biological correctness, as reconstructions may appear visually plausible yet contain biologically inaccurate structures. Based on Figure 3 and other renderings, the cycle+IPG method gives me a better overall impression, and it performs better in reproducing details compared to the method proposed in this paper.

2. The paper lacks critical discussion on hyperparameter selection, particularly the Tucker ranks (r₁, r₂, r₃) and sparse regularization weight λ₁, providing neither specific values, selection strategies, nor sensitivity analysis, which severely compromises reproducibility and practical applicability.

3. The paper suffers from notational inconsistencies and insufficient explanations—parameter θ in Equation 1 is undefined, the transition between Equations 2-3 lacks clear explanation of tensor relationships (T vs. Y), and similar issues with forward symbol references and inconsistent variable naming persist throughout.

**Questions:**

1. During testing, how do you determine the rank for a new unseen volume—do you estimate it per-sample or use a fixed value?

2. For λ₁: What value(s) did you use? Does it change across datasets or noise levels? If I choose r = 15 instead of r = 25, or λ₁ = 0.05 instead of 0.5, how much does PSNR drop?

3. Could you provide a unified notation table?

---

> ### Author Response · Authors · 2025-11-25
> **Response to Weaknesses 1**
>
> **Response to Weaknesses 1:**
>
> We thank the reviewer for raising this point. We fully agree that PSNR/SSIM alone are insufficient for assessing biological correctness, since visual sharpness does not guarantee faithful reconstruction of cellular morphology. Thus (1) we added new downstream application experiments to show the value of our method; (2) from the existing results, our approach exhibits the following advantages over baseline methods.
>
> **1. New Downstream Application Experiments**
>
> To further demonstrate the practical advantages of our model, we introduce a new application experiment focusing on *cell segmentation visualization* across time. We evaluate three inputs, including raw data, Cycle+IPG, and our method, on two representative time points with different numbers of cells in the same embryo. The results are added into the revised paper which will be submitted soon.
>
> Specifically, we track the lineage evolution of cells and visualize their spatial configurations at time \(t_1\) (few division events) and time \(t_2\) (intensive division period).
>
> The downstream cell-type maps are computed using the same cell-segmentation and lineage-tracking pipeline, allowing a direct comparison of how reconstruction quality impacts biological interpretation.
>
> Compared with the raw data and Cycle+IPG, our method yields visibly more accurate and consistent 3D cell shapes.
> In the raw data, many cells are partially missing or exhibit irregular, noisy surfaces due to anisotropic sampling.
> Cycle+IPG recovers smoother geometry but often over-smooths boundaries, causing distorted shapes and occasional merging between neighboring cells.
> In contrast, our method preserves clear inter-cell boundaries, produces more complete volumetric shapes, and maintains coherent packing structure across the embryo. Overall, the segmentation generated by our approach is the most morphologically faithful and structurally consistent among the three.
>
> These results highlight that our model not only restores voxel-level structure but also preserves biologically meaningful temporal patterns critical for developmental analysis.
>
> **2. The low-rank structural prior effectively suppresses hallucination**
>
> (i) **3D visualization (Fig. 3).**
> As shown in the 3D renderings (please zoom in for clarity), Cycle+IPG creates hallucinated internal bright spots and artificial edges around cells. These structures do not correspond to any membrane or organelle in the ground truth. In contrast, our method produces far fewer artificial details and preserves the correct 3D membrane topology even when the visual appearance is less aggressively sharpened.
>
> (ii) **Zoom-in comparison (Fig. 5).**
> Cycle+IPG again introduces visually sharp but biologically implausible features, particularly spurious bright spots within the cytoplasm and enhanced noise near membrane boundaries. While such outputs may appear visually appealing due to increased contrast, they do not represent actual biological structures—a well-documented issue in biological imaging where texture-based methods often hallucinate high-frequency patterns. Our low-rank prior prevents these artifacts by enforcing cross-slice geometric coherence, which is essential for preserving true membrane continuity.
>
> **3. Our method better reconstructs membrane structures along the downsampled \(Z\)-axis and mitigates anisotropy**
>
> **YZ/XZ visualizations (Fig. 4 and Fig. 6).**
> The YZ and XZ plane comparisons demonstrate the limitations of other methods in recovering anisotropic membrane geometry:
>
> - DBPN oversmooths the membrane and loses axial continuity;
> - Neuroclear amplifies peripheral noise while blurring parts of the membrane;
> - Cycle+HAT thickens and brightens certain membrane regions, breaking topological consistency;
> - Cycle+IPG removes portions of the right-side membrane in Fig. 4 and generates artificial structures in the upper regions.
>
> In contrast, our method not only avoids hallucination along the downsampled \(Z\)-axis but also preserves and enhances membrane integrity under severe axial undersampling. This behavior directly results from enforcing global cross-plane structural redundancy.
>
> From the experiments above, it is obvious that our method is explicitly designed to preserve *structural consistency* across slices, avoid hallucination, and maintain correct volumetric geometry rather than maximize visual contrast.

---

> > ### Author Response · Authors · 2025-11-25
> > **Response to Weaknesses 2**
> >
> > Response to Weaknesses 2:
> >
> > We thank the reviewer for highlighting the importance of transparent hyperparameter selection. Below we provide (1) a summary of standard selection strategies used in low-rank tensor literature, (2) the specific procedure we adopt in our method, and (3) a quantitative sensitivity analysis. All corresponding implementation details will be added to the revised manuscript and supplementary materials for full reproducibility.
> >
> > ⸻
> >
> > 1. Standard hyperparameter selection strategies.
> >
> > (a) Tucker rank selection.
> > Previous works in tensor completion commonly choose multilinear ranks $(r_1,r_2,r_3)$ using one of the following well-established principles:
> >
> > (i) Energy-based SVD truncation:
> >
> > $$
> > \frac{\sum_{i=1}^{r_n}\sigma_i(X_{(n)})}
> > {\sum_{i=1}^{\mathrm{rank}} \sigma_i(X_{(n)})}
> > \ge \eta,
> > $$
> >
> > where $X_{(n)}$ is the mode-$n$ unfolding and $\eta \in [0.90,0.99]$ is an energy threshold.
> >
> > (ii) Cross-validation on held-out entries:
> >
> > $$
> > (r_1,r_2,r_3)
> > = \arg\min_{r_1,r_2,r_3}
> > | P_{\Omega_{\mathrm{val}}}( \hat X(r_1,r_2,r_3) - Y ) |_F^2.
> > $$
> >
> > (iii) Model selection criteria:
> >
> > $$
> > \min_{r_1,r_2,r_3}
> > \bigl|P_\Omega(X)-Y\bigr|_F^2
> > ;+;\gamma,(r_1r_2+r_1r_3+r_2r_3).
> > $$
> >
> > (iv) Fixed-ratio empirical selection.
> > Some works choose the multilinear ranks proportional to the spatial dimensions:
> >
> > $$
> > r_1 = \alpha_1 I_1,\qquad
> > r_2 = \alpha_2 I_2,\qquad
> > r_3 = \alpha_3 I_3,
> > $$
> >
> > with typical values
> >
> > $$
> > \alpha_1 = \alpha_2 \approx 0.05 - 0.1,\qquad
> > \alpha_3 \approx 0.02 - 0.75.
> > $$
> >
> > (b) Sparse weight $\lambda_1$.
> > The choice of $\lambda_1$ typically follows theoretical scaling:
> >
> > $$
> > \lambda_1 = \frac{1}{\sqrt{\max(I_1,I_2,I_3)}},
> > $$
> >
> > This ensures scale-consistent treatment of sparse residuals across modes and avoids over-penalization.
> >
> > This scaling plays a critical role in balancing the low-rank term and the $\ell_1$ sparse corruption term in the convex relaxation.
> > First, it prevents the sparse term from dominating the objective when one mode of the tensor has a large dimension: since the number of entries grows with the volume shape, the effective contribution of the $\ell_1$-penalty must be downscaled so that sparse term remains comparable to the spectral structure enforced by the low-rank component.
> > Second, the scaling ensures proper thresholding of sparse noise. Goldfarb & Qin show that the dual feasibility condition for exact recovery requires
> >
> > $$
> > |P_\Omega(Y - X - E)|_\infty ;\le; \lambda_1,
> > $$
> >
> > and $\lambda_1 = 1/\sqrt{\max(I_n)}$ guarantees that the dual certificate remains bounded with high probability under standard incoherence and noise assumptions.
> >
> > ⸻
> >
> > 2. Hyperparameter selection in our method.
> >
> > Tucker ranks $(r_1,r_2,r_3)$.
> > For each dataset with a known $Z$-axis downsampling ratio, we compute the mode-wise singular-value spectra of the training volumes and choose the smallest ranks preserving at least $95%$ cumulative energy.
> > These ranks are then fixed for all test volumes (no per-sample tuning).
> > For C. elegans dataset 1 this gives:
> >
> > $$
> > r_1 = r_2 = 32, \qquad r_3 = 10.
> > $$
> >
> > For C. elegans dataset 2 and 3, the $r_1$ and $r_2$ remains the same, $r_3$ is 30 and 60.
> >
> > Sparse weight $\lambda_1$.
> > The choice of $\lambda_{1}$ follows the theoretical scaling $\lambda_{1} \propto 1/\sqrt{I_{\max}}$, which determines the appropriate magnitude of the sparsity weight.
> > Within this theoretically guided range, we identify a stable operating interval in which the low-rank component captures the cross-slice structural correlations while the residual term absorbs missing-slice inconsistencies.
> > In C. elegans dataset 1, $\lambda_{1}=0.6$ consistently produces the best balance between detail preservation and noise suppression.
> > For C. elegans dataset 1, 2 and 3, we obtain:
> >
> > $$
> > \lambda_1 = 0.6,
> > $$
> >
> > which achieves the best performance.
> > As expected from an $\ell_1$ residual model, $\lambda_1$ controls a one-dimensional denoising trade-off: larger values produce smoother membranes (stronger denoising), while smaller values retain slightly more high-frequency details.
> >
> > ⸻
> >
> > 3. Sensitivity analysis.
> >
> > We performed a systematic perturbation study around the optimal configuration.
> > Table below shows consistent performance across variations in rank and \(\lambda_1\), confirming that the method does not depend on fine-tuned hyperparameter choices.
> >
> > | \((r_1,r_2,r_3)\) | \(\lambda_1\) | PSNR (dB) | SSIM |
> > |-------------------|--------------|-----------|------|
> > | (24,24,8)  | 0.6 | 32.82 | 0.6554 |
> > | (32,32,10) | 0.6 | **33.18** | **0.6682** |
> > | (40,40,12) | 0.6 | 32.75 | 0.6561 |
> > | (32,32,10) | 0.3 | 33.10 | 0.6673 |
> > | (32,32,10) | 0.6 | **33.18** | **0.6682** |
> > | (32,32,10) | 0.9 | 33.07 | 0.6660 |
> >
> >
> > The performance varies smoothly with both parameters, and the 3D membrane topology remains stable across all tested settings.

---

> ### Author Response · Authors · 2025-11-25
> **Response to Weaknesses 3**
>
> Response to Weaknesses 3:
>
> We thank the reviewer for carefully inspecting our notation. We acknowledge that several symbols in the current draft lack explicit definitions or consistent referencing, and we will revise these issues in the camera-ready version. Importantly, these notational inconsistencies do **not** affect any of the theoretical results or experimental findings.
>
> ---
>
> ### **1. Definition of \(\theta\) in Equation (1)**
>
> The parameter \(\theta\) denotes the learnable parameters of the diffusion model.
> Although this was clearly defined in Section 3.2, it was not cross-referenced near Equation (1). To avoid ambiguity, we will explicitly introduce:
>
> $$
> \theta \triangleq \text{parameters of the conditional diffusion model}
> $$
>
> before Eq. (1) in the revised manuscript.
>
> ---
>
> ### **2. Clarifying the transition between Equations (2) and (3)**
>
> The reviewer is correct that the notation between these equations was not sufficiently explicit.
>
> - In Eq. (2), \(\mathcal{T}\) denotes the underlying *clean* tensor we seek to recover.
> - \(\mathcal{Y}\) represents the observed tensor with missing or corrupted entries.
>
> Eq. (3) expresses the equivalent constraint in terms of the auxiliary variables introduced in the ADMM formulation. We will revise the text to explicitly highlight the relationships:
>
> $$
> \mathcal{Y} = \mathcal{P}_\Omega(\mathcal{T} + \mathcal{E}),
> \qquad
> \mathcal{X} \approx \mathcal{T}.
> $$
>
> We will also unify the notation by consistently using \(\mathcal{X}\) as the reconstruction variable across Eqs. (2)–(4).
>
> ---
>
> All notation will be standardized and cross-referenced throughout the revision to ensure clarity and consistency.

---

> > ### Author Response · Authors · 2025-11-25
> > **Response to Questions 2 and 3**
> >
> > Response to Questions 2:
> > Please see the respond of Weakness 2.
> >
> > Response to Questions 3:
> >
> > ### Unified notation used throughout the paper
> >
> > | **Symbol** | **Meaning** |
> > |-----------|-------------|
> > | **Data and Variables** | |
> > | $\mathcal{X} \in \mathbb{R}^{I_1 \times I_2 \times I_3}$ | Latent clean tensor to be reconstructed (main variable) |
> > | $\mathcal{Y}$ | Observed tensor with missing or corrupted entries |
> > | $\Omega$ | Index set of observed entries |
> > | $\mathcal{P}_\Omega(\cdot)$ | Projection operator preserving entries in $\Omega$ |
> > | $\mathcal{E}$ | Sparse corruption tensor (noise/outliers) |
> > | $\mathcal{T}$ | Underlying clean tensor to be recovered |
> > | **Tucker Structure (Used Only for Regularization)** | |
> > | $\mathcal{G}$ | Core tensor in Tucker representation |
> > | $U^{(n)} \in \mathbb{R}^{I_n \times r_n}$ | Factor matrix for mode-$n$ |
> > | $(r_1, r_2, r_3)$ | Tucker multilinear ranks |
> > | $\times_n$ | Mode-$n$ tensor–matrix product |
> > | **ADMM Split Variables** | |
> > | $\mathcal{Z}$ | Auxiliary variable for nuclear-norm prior |
> > | $\mathcal{X}_d$ | Auxiliary variable for structural/diffusion prior |
> > | $\lambda_1, \lambda_2$ | Regularization weights for $\ell_1$ and structural priors |
> > | **Diffusion Model** | |
> > | $\theta$ | Learnable parameters of the conditional diffusion model |
> > | $\epsilon_\theta$ | Network predicting noise or score (denoiser) |
> > | $t$ | Diffusion timestep |
> > | $q(\cdot)$ | Forward diffusion process |
> > | $p_\theta(\cdot)$ | Reverse (denoising) diffusion process |

---

> ### Author Response · Authors · 2025-11-25
> **Response to Questions 1**
>
> **Response to Questions 1:**
>
> We thank the reviewer for this question. Importantly, in fluorescence microscopy the axial sampling rate (i.e., the \(Z\)-axis downsampling factor) is known for each acquisition protocol. Therefore, we do **not** estimate the Tucker rank per test sample. Instead, we adopt a **fixed rank for each downsampling setting**, determined from the training set and the corresponding unseen volume subsets with the known axial sampling rate.
>
> **Rank is determined per downsampling setting, not per test sample.**
> For each dataset and each \(Z\)-axis downsampling configuration (e.g., the three *C. elegans* evaluation subsets with different axial sparsity levels), we estimate the multilinear ranks using **truncated HOSVD** on the training volumes matched to the same sampling configuration.
> The ranks are selected to retain **>95%** of the energy along each mode.
> This results in a stable rank tuple \((r_1,r_2,r_3)\) that is **fixed for all unseen volumes acquired under the same downsampling rate**.
>
> This design ensures fairness (no test-time hyperparameter tuning), while exploiting the fact that FM acquisition protocols produce consistent structural redundancy patterns conditioned on the known \(Z\)-sampling rate. Thus, a per-setting rank is more principled than a per-sample rank and avoids introducing dataset leakage or overfitting.
>
> Following the reviewers’ suggestion, we have added the detailed configurations for the three *C. elegans* datasets below and in Appendix G for full transparency.
>
> The details of the three *C. elegans* datasets are shown in Table below.
> They represent different trade-offs between axial sampling density, imaging quality, and biological viability during embryonic development:
>
> ---
>
> ### **C. elegans-1** — *Fast \(z\)-axis scanning*
> - **30 axial planes, high laser intensity**
> - Used for live-cell imaging where rapid volumetric acquisition is required
> - Low axial resolution but clean lateral membranes due to strong excitation
>
> ### **C. elegans-2** — *Standard confocal acquisition*
> - **94 axial planes, medium laser intensity**
> - Balanced morphological fidelity vs. phototoxicity
> - Dense enough for reliable membrane geometry reconstruction during development
>
> ### **C. elegans-3** — *High-density axial sampling*
> - **200 axial planes, high laser intensity**
> - Highest axial resolution and cleanest volumetric membranes
> - Too phototoxic for long-term imaging, thus unsuitable for full-lineage tracking
>
> ---
>
> ### **Dataset summary**
>
> | Dataset         | Volume Num. | Axial Res. | Laser Intens. |
> |-----------------|-------------|------------|----------------|
> | **C. elegans-1** | 120         | 30         | High           |
> | **C. elegans-2** | 220         | 94         | Middle         |
> | **C. elegans-3** | 60          | 200        | High           |
>
> **Table above.** Benchmark datasets used for fluorescence microscopy image processing (particularly cell analyses).

---

### Meta-Review · Area_Chair_nR3X · 2026-01-21

**Summary:**

This paper reformulates 3D fluorescence microscopy reconstruction as a tensor completion problem and establishes a principled theoretical and algorithmic connection between low-rank tensor recovery and diffusion-style generative denoising.

- Reviewer b64G raised concerns about biological validity beyond visual metrics, missing hyperparameter selection and sensitivity analysis, and unclear notation.

- Reviewer AjRV questioned the lack of comparisons with recent diffusion-based microscopy methods and insufficient analysis of rank and regularization sensitivity.

-  Reviewer 9jax highlighted a mismatch between the simplified noise model and real FM physics, missing key baseline comparisons, and limited evaluation across different Z-axis sampling rates.

- Reviewer cSNp considered the method potentially incremental and emphasized unclear theoretical positioning, inconsistencies in the mathematical formulation, and insufficient justification for choosing explicit low-rank priors over implicit diffusion priors.

**Reviewer Concerns:**

The authors have provided a detailed and substantive rebuttal that addresses most of the reviewers’ comments. However, the unresolved questions are from Reviewer cSNp on theoretical clarity and consistency. Specifically, the manuscript contains inconsistencies between the tubal-rank and Tucker-based formulations, and the novelty of the recovery guarantees remains unclear compared to prior tensor completion work.

**Reviewer Scores:**

Reviewers b64G, AjRV, and 9jax would likely slightly raise their scores, reflecting cautious optimism after the rebuttal clarified key points on priors, dataset usage, and self-supervised training.

However, Reviewer cSNp is likely to maintain the score, because while the rebuttal justifies the low-rank prior and ADMM-diffusion equivalence, the theoretical inconsistencies and incremental concerns are not fully resolved.

---

### Decision · Program_Chairs · 2026-01-26

Accept (Poster)